# Global Downregulation of Penicillin Resistance and Biofilm Formation by MRSA Is Associated with the Interaction between Kaempferol Rhamnosides and Quercetin

Xinlong He,[a,c,e,f,g] Wenwen Zhang,[a] Qingchao Cao,[b] Yinyue Li,[a] Guangyu Bao,[c] Tao Lin,[c] Jiaojiao Bao,[c] Caiwang Chang,[c] Changshui Yang,[d] Yi Yin,[a] Jiahui Xu,[a] Zhenyu Ren,[a] Yingshan Jin,[b,e] Feng Lu[a,c,e,f,g]

[a]Department of Pathogenic Biology, School of Medicine, Yangzhou University, Yangzhou, People's Republic of China
[b]College of Bioscience and Biotechnology, Yangzhou University, Yangzhou, People's Republic of China
[c]Affiliated Hospital of Yangzhou University, Yangzhou, People's Republic of China
[d]Department of Pharmacy, School of Medicine, Yangzhou University, Yangzhou, People's Republic of China
[e]Jiangsu Key Laboratory of Zoonosis, Yangzhou University, Yangzhou, People's Republic of China
[f]Jiangsu Key Laboratory of Experimental & Translational Non-coding RNA Research, School of Medicine, Yangzhou University, Yangzhou, People's Republic of China
[g]Jiangsu Co-Innovation Center for the Prevention and Control of Important Animal Infectious Diseases and Zoonoses, College of Veterinary Medicine, Yangzhou University, Yangzhou, People's Republic of China

Xinlong He, Wenwen Zhang, and Qingchao Cao contributed equally to this work. The order of authors was determined according to the research collaboration agreement reached in advance.

**ABSTRACT** The rapid development of methicillin-resistant *Staphylococcus aureus* (MRSA) drug resistance and the formation of biofilms seriously challenge the clinical application of classic antibiotics. Extracts of the traditional herb *Chenopodium ambrosioides* L. were found to have strong antibiofilm activity against MRSA, but their mechanism of action remains poorly understood. This study was designed to investigate the antibacterial and antibiofilm activities against MRSA of flavonoids identified from *C. ambrosioides* L. in combination with classic antibiotics, including ceftazidime, erythromycin, levofloxacin, penicillin G, and vancomycin. Liquid chromatography-mass spectrometry (LC-MS) was used to analyze the nonvolatile chemical compositions. Reverse transcription (RT)-PCR was used to investigate potential multitargets of flavonoids based on global transcriptional responses of virulence and antibiotic resistance. A synergistic antibacterial and biofilm-inhibiting activity of the alcoholic extract of the ear of *C. ambrosioides* L. in combination with penicillin G was observed against MRSA, which proved to be closely related to the interaction of the main components of kaempferol rhamnosides with quercetin. In regard to the mechanism, the increased sensitivity of MRSA to penicillin G was shown to be related to the downregulation of penicillinase with SarA as a potential drug target, while the antibiofilm activity was mainly related to downregulation of various virulence factors involved in the initial and mature stages of biofilm development, with SarA and/or $\sigma$B as drug targets. This study provides a theoretical basis for further exploration of the medicinal activity of kaempferol rhamnosides and quercetin and their application in combination with penicillin G against MRSA biofilm infection.

**IMPORTANCE** In this study, the synergistic antibacterial and antibiofilm effects of the traditional herb *C. ambrosioides* L. and the classic antibiotic penicillin G on MRSA provide a potential strategy to deal with the rapid development of MRSA antibiotic resistance. This study also provides a theoretical basis for further optimizing the combined effect of kaempferol rhamnosides, quercetin, and penicillin G and exploring anti-MRSA biofilm infection research with SarA and $\sigma$B as drug targets.

Address correspondence to Feng Lu, lufeng@yzu.edu.cn, or Yingshan Jin, ysjin@yzu.edu.cn.

The authors declare no conflict of interest.

**KEYWORDS** *Staphylococcus aureus*, MRSA, biofilm formation, antibiotic resistance, global regulation, *Chenopodium ambrosioides* L., kaempferol rhamnosides, quercetin

Staphylococcus aureus, the most common pathogen in human infection, can cause local purulent infection, pneumonia, pseudomembranous colitis, pericarditis, and even sepsis (1). Soon after penicillin was introduced for therapeutic use in 1940s, an increased incidence of penicillin resistance was reported in *S. aureus* strains (2). Methicillin, a semisynthetic penicillin, is resistant to penicillinase, but unfortunately, methicillin-resistant *S. aureus* (MRSA) emerged shortly after its clinical application (3). Today, MRSA is the leading cause of nosocomial infections (4). The prevalence of MRSA infection has resulted in the marginalization of the clinical use of traditional classic antibiotics. On the other hand, the current development of new drugs is lagging seriously behind the rapid development of drug resistance, and the emergence of completely drug-resistant superbugs is bound to put patients in an incurable dilemma. Repurposing of old drugs with the idea of reducing toxicity is now considered a potentially effective strategy to control MRSA infection (5, 6).

Biofilms are microbial communities embedded in an extracellular matrix (ECM) composed of lipids, proteins, polysaccharides, and DNA that can form on medical implants or tissue surfaces to protect pathogenic bacteria from immune clearance and antibiotic killing and are therefore implicated in most chronic infections (7, 8). Biofilms are formed in a multistage process that includes initial attachment, accumulation, maturation, and dispersion, and various virulence factors are involved in the progress along the stages of *S. aureus* biofilm formation (9). For example, surface protein adhesins, typically known as MSCRAMMs (microbial surface components recognizing adhesive matrix molecules), including fibrinogen-binding proteins (FnBPs), fibrinogen-binding clumping factors (Clfs), elastin-binding protein (EbpS), and autolysin (Atl), confer the ability for *S. aureus* to adhere to the host matrix (10). Polysaccharide intercellular adhesin (PIA), also referred as poly-*N*-acetylglucosamine (PNAG), contributes to biofilm accumulation (11). Extracellular DNA (eDNA) released by bacterial autolysis can be an important component of the biofilm matrix (12). Proteases and phenol-soluble modulin (PSM) peptides may act as dispersants of biofilm to aid in bacterial dissemination and the settlement of new biofilms at a distant site (13). Moreover, the coordinated expression of these virulence factors is modulated in complex networks that include the global regulators accessory gene regulator (Agr), staphylococcal accessory regulator (SarA), and alternative sigma factor B ($\sigma$B) (14). Therefore, these virulence factors and their regulators are potential targets for effective prevention and control of *S. aureus* biofilm-associated infections.

Flavonoids are one of a large class of plant medicinal components with antioxidant, anti-inflammatory, antiallergic, anticancer, antiviral, and antifungal properties (15). Recent studies have shown that, in addition to antibacterial effects, some flavonoids also inhibit bacterial biofilm formation by affecting bacterial adhesion, motility, and quorum sensing (QS) (16). *Chenopodium ambrosioides* L., an annual or perennial herb, is widely distributed on the planet, and its derived essential oil is commonly used in folk medicine as an antirheumatic, anti-inflammatory, antipyretic, anthelmintic, antifungal, and antiulcer agent (17–21). However, the biological activities and mechanisms of action in terms of antibacterial and antibiofilm activities remain poorly understood. In the previous stage of our research, we compared the antibiofilm activities of ethanol extracts (EE) and water extracts (WE) from the root (GR), stem (GS), and ear (GE) of *C. ambrosioides* L. originating from Guangxi, China, and found that GE-EE had the most significant inhibitory effect on biofilm formation by *S. aureus* strain ATCC 43300. It was preliminarily estimated that flavonoids were the main active components. The purpose of this study was to investigate the antibacterial and antibiofilm activities against MRSA of flavonoids identified from *C. ambrosioides* L. in combination with traditional antibiotics, including ceftazidime, erythromycin, levofloxacin, penicillin G, and vancomycin, and to investigate potential multitargets of flavonoids based on global transcriptional responses of virulence and antibiotic resistance.

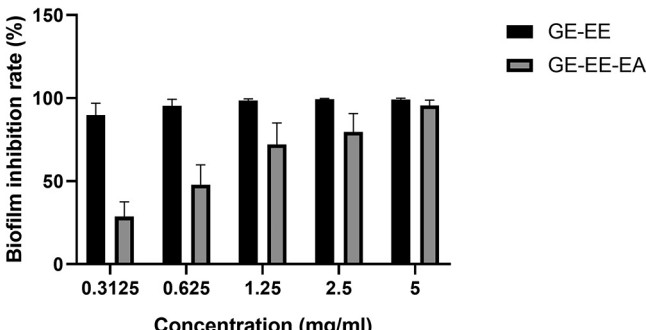

**FIG 1** Antibiofilm activities of different extracts from *C. ambrosioides* L. GE, ear of *C. ambrosioides* L. collected from Guangxi, China; EE, ethanol extracts; EA, fractional extraction with ethyl acetate. All data are presented as the mean values from three biological replicates. Error bars show SD.

## RESULTS AND DISCUSSION

**Quercetin optimized biofilm inhibition by kaempferol glycosides against *S. aureus*.** Significant inhibitory activity against biofilm formation was observed for all extracts of *C. ambrosioides* L. (Fig. S1 in the supplemental material), but no significant inhibitory activity against the growth of *S. aureus* ATCC 43300 (Fig. S2). It was observed that, among the 6 extracts of GS, GR, and GE extracted by the immersion method using water (WE) and ethanol (EE) as the solvents, GE-EE exhibited relatively high biofilm inhibition activity against *S. aureus* ATCC 43300, while among the fractionated extracts of GE-EE, GE-EE-EA (extracted with ethyl acetate) showed the highest biofilm inhibition activity (Fig. S1). However, the antibiofilm activity of GE-EE was significantly higher ($P < 0.05$) than that observed for GE-EE-EA at concentrations ranging from 0.3125 to 2.5 mg/mL (Fig. 1). This could be due to chemical changes in various kaempferol glycoside derivatives, as well as other flavonoids, as shown in Table 1. The relative contents of naringin dihydrochalcone, kaempferol-3-rutinoside, naringin, rutin, and kaempferol-3,7-dirhamnoside were significantly increased ($P < 0.05$) after fractionated extraction with ethyl acetate in GE-EE, while the relative contents of quercetin and kaempferol-7-*O*-rhamnoside were significantly decreased ($P < 0.05$). Kaempferol glycosides are common bioactive components in *Chenopodium* (22). The cumulative proportion of kaempferol glycosides in GE-EE was up to 67.5%, suggesting that it is likely to be the main active component for biofilm inhibition.

Furthermore, GE-EE ($>62.5$ $\mu$g/mL) was found to effectively inhibit biofilm formation by *S. aureus* BWSA11 (Fig. 2), a clinical MRSA strain with robust biofilm-forming

**TABLE 1** Relative contents of compounds identified from ethanol extracts of *C. ambrosioides* L.

| Retention time (min) | Mol wt (*m/z*) | Compound | Relative proportion (%) in[c]: | |
|---|---|---|---|---|
| | | | GE-EE | GE-EE-EA |
| 19.013 | 582 | Naringin dihydrochalcone[a] | 0.00 | 3.72* |
| 20.018 | 302 | Quercetin[a] | 2.92 | 0.00* |
| 20.587 | 594 | Kaempferol-3-rutinoside[a,b] | 0.00 | 2.81* |
| 21.113 | 580 | Naringin[a] | 0.00 | 1.32* |
| 21.333 | 610 | Rutin[a] | 0.00 | 0.79* |
| 24.147 | 578 | Kaempferol-3,7-dirhamnoside[a,b] | 14.25 | 33.55* |
| 24.905 | 564 | Kaempferol-3-*O*-apigenin-7-*O*-rhamnoside[a,b] | 24.17 | 34.04 |
| 26.745 | 434 | Kaempferol-7-*O*-rhamnoside[a,b] | 2.69 | 0.00* |
| 31.861 | 418 | Syringaresinol | 29.59 | 1.13* |
| 33.578 | 606 | Kaempferol-3-*O*-acetylapigenin-7-*O*-rhamnoside[a,b] | 26.39 | 22.64 |

[a]Flavonoid compound.
[b]Kaempferol glycoside.
[c]GE, the ear of *C. ambrosioides* L. collected from Guangxi, China; EE, ethanol extracts; EA, fractional extraction with ethyl acetate; *, significant at the level of 0.05 (2-tailed).

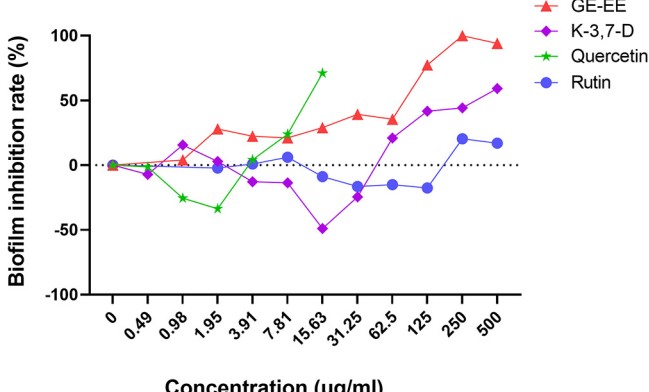

**FIG 2** Comparison of antibiofilm activities between GE-EE and representative components. GE, ear of *C. ambrosioides* L. collected from Guangxi, China; EE, ethanol extracts; K-3,7-D, kaempferol-3,7-dirhamnoside. All data are presented as the mean values from three biological replicates.

ability (23). The biofilm-inhibitory activity of GE-EE was further found to be significantly correlated with kaempferol-3,7-dirhamnoside, a representative kaempferol glycoside, but not with quercetin and rutin (Table 2). This result is in agreement with a previous finding that kaempferol inhibited the primary attachment phase of biofilm formation in *S. aureus* (24). It was further found that quercetin (1.95 to 7.81 $\mu$g/mL) and kaempferol-3,7-dirhamnoside (15.63 to 125 $\mu$g/mL) had a synergistic inhibitory effect on *S. aureus* BWSA11 biofilm (Fig. 3), while rutin and kaempferol-3,7-dirhamnoside did not (data not shown). This finding could explain the higher antibiofilm activity (Fig. S1) but lower total content of kaempferol glycosides in GE-EE (Table 1). Subinhibitory concentrations of antibiotics can induce the formation of *S. aureus* biofilms (25); consistent with this, quercetin (1.95 $\mu$g/mL) and kaempferol-3,7-dirhamnoside (15.63 $\mu$g/mL) also induced the formation of *S. aureus* strain BWSA11 biofilms (Fig. 2). Unlike quercetin or kaempferol-3,7-dirhamnoside, GE-EE showed no *S. aureus* biofilm-inducing activity at all concentrations tested. It was found that the biofilm-inhibitory activities of quercetin and kaempferol-3,7-dirhamnoside within a range of concentrations (0.49 to 15.63 $\mu$g/mL) against *S. aureus* BWSA11 exhibited opposite changes (Fig. 2) and were inversely correlated (Table 2), which was likely to result in some degree of mutual containment in antibiofilm or biofilm-inducing activity. This might explain the biofilm-inhibitory properties of GE-EE that were different from those of single active ingredients. Therefore, an appropriate combination of active ingredients, such as quercetin and kaempferol glycosides, may be effective in eliminating the risk of biofilm induction posed by inappropriate application of the single ingredients.

**Synergistic antibacterial and antibiofilm activity of nonmonomeric kaempferol glycosides in combination with penicillin G against *S. aureus*.** It was observed that,

**TABLE 2** Correlation matrix of Pearson correlation coefficients between compounds in antibiofilm and antibacterial activities against *S. aureus* strain BWSA11

| | PCC for indicated activity[a] | | | | | | | |
|---|---|---|---|---|---|---|---|---|
| | Antibiofilm | | | | Antibacterial | | | |
| Factor | GE-EE | K-3,7-D | Quercetin | Rutin | GE-EE | K-3,7-D | Quercetin | Rutin |
| GE-EE | 1 | 0.718* | 0.461 | 0.474 | 1 | 0.799** | 0.523 | 0.836** |
| K-3,7-D | | 1 | −0.954** | 0.417 | | 1 | −0.606 | 0.495 |
| Quercetin | | | 1 | −0.414 | | | 1 | 0.948** |
| Rutin | | | | 1 | | | | 1 |

[a]GE, ear of *C. ambrosioides* L. collected from Guangxi, China; EE, ethanol extracts; K-3,7-D, kaempferol-3,7-dirhamnoside. Pearson correlation coefficients (PCCs) were calculated for strain BWSA11 exposed to quercetin (0.49 to 15.63 $\mu$g/mL), GE-EE (0.98 to 500 $\mu$g/mL), K-3,7-D (0.49 to 500 $\mu$g/mL), and rutin (1.95 to 500 $\mu$g/mL). **, significant at the level of 0.01 (2-tailed); *, significant at the level of 0.05 (2-tailed).

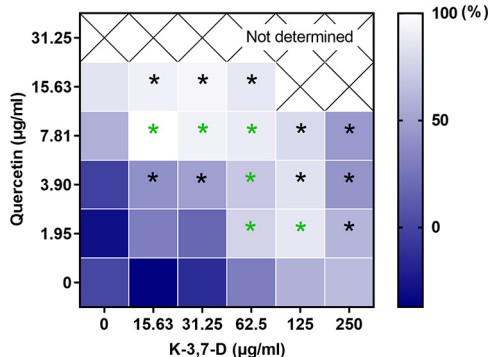

**FIG 3** Heat map of antibiofilm effects of quercetin in combination with kaempferol glycoside. K-3, 7-D, kaempferol-3,7-dirhamnoside. Not determined, the biofilm formation of the area where the bacterial inhibition rate was over 95% was not calculated. Black asterisks indicate significant decreases at a *P* value of 0.05 compared to the control. Green asterisks indicate significant decreases at a *P* value of 0.05 compared to the control and either of the two parallel single-agent groups. All data are presented as the mean values from three biological replicates.

with the exception of quercetin (>7.81 $\mu$g/mL), neither GE-EE, kaempferol-3,7-dirhamnoside, nor rutin significantly inhibited the growth of *S. aureus* BWSA11 (Fig. 4). This observation was in agreement with the result that kaempferol had less bacteriostatic activity against *S. aureus* (26). Considering the extremely low content of quercetin in GE-EE, it is not difficult to explain why the antibacterial property of GE-EE was closely related to those of kaempferol glycosides (Table 2). Limited by the solubility of kaempferol-3,7-dirhamnoside, it is difficult to determine whether there was a synergistic or additive antibacterial effect between quercetin and kaempferol-3,7-dirhamnoside against the growth of *S. aureus* BWSA11 (fractional inhibitory concentration [FIC] of <1.5). However, as shown by the results in Fig. 5, the presence of kaempferol-3,7-dirhamnoside (125 and 250 $\mu$g/mL) and quercetin (15.63 $\mu$g/mL) reduced each other's MIC values against *S. aureus* BWSA11 by 2-fold and at least 8-fold. Therefore, a proper compatibility of kaempferol glycosides and quercetin is beneficial to the mutual promotion of their antibacterial activities. Furthermore, GE-EE in combination with penicillin G was found to synergistically inhibit the growth of strains BWSA11, BWSA15, and ATCC 43300 with FICs of <0.19, <0.5, and <0.16, respectively (Fig. 5). Interestingly, neither the combination of quercetin and penicillin G nor the combination of kaempferol-3,7-dirhamnoside and penicillin G appeared to synergistically inhibit the growth of representative strains tested in this study (data not shown). This is in agreement with a previous finding that kaempferol and quercetin had mild inhibitory effects on

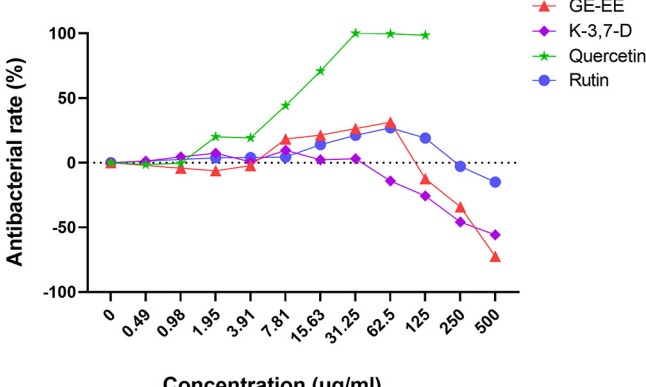

**FIG 4** Comparison of bacterial-inhibition activities between GE-EE and representative components. GE, ear of *C. ambrosioides* L. collected from Guangxi, China; EE, ethanol extracts; K-3,7-D, kaempferol-3, 7-dirhamnoside. All data are presented as the mean values from three biological replicates.

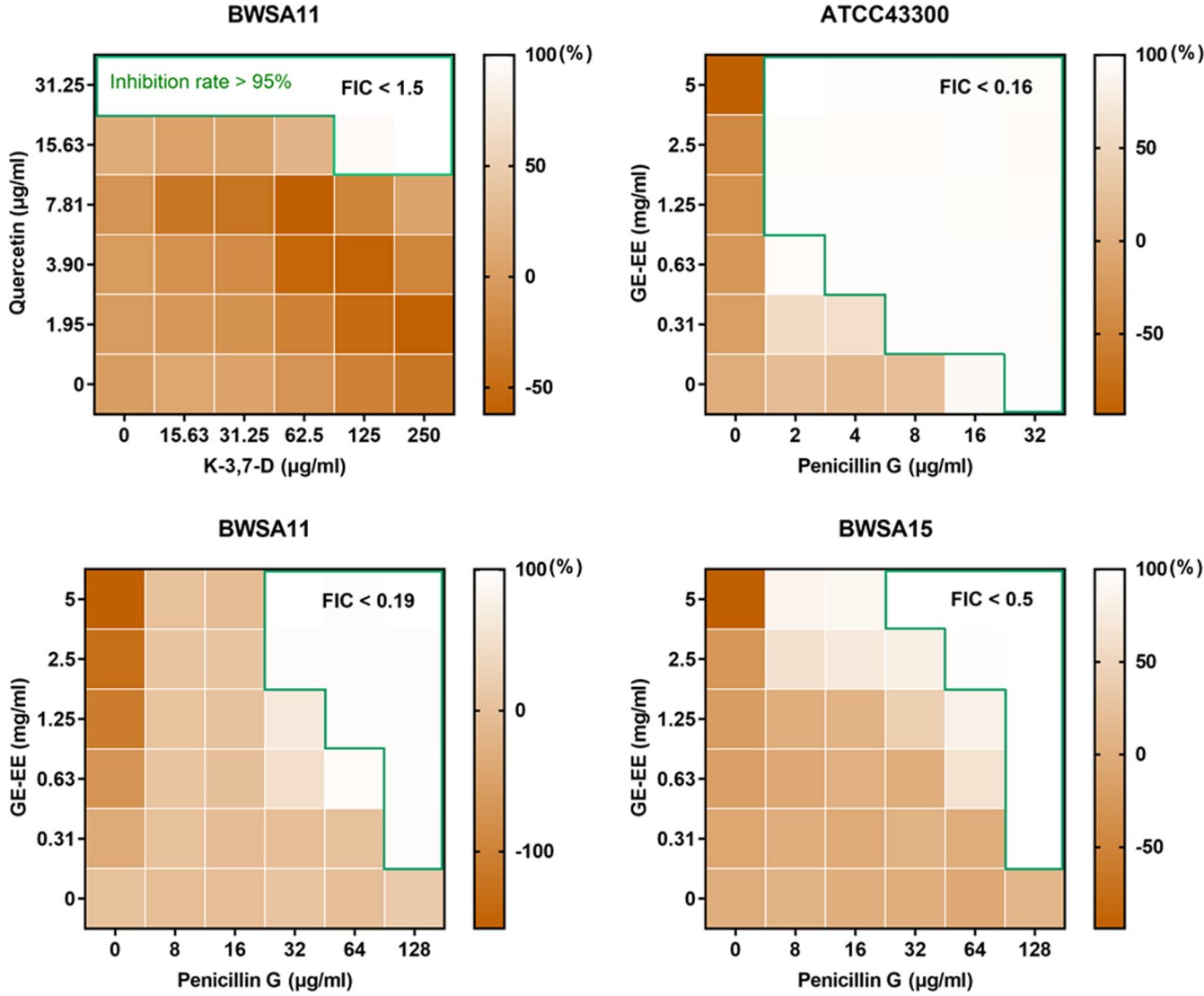

**FIG 5** Heat map of antibacterial effects of combinations between penicillin G and representative components. GE, ear of *C. ambrosioides* L. collected from Guangxi, China; EE, ethanol extracts; K-3,7-D, kaempferol-3,7-dirhamnoside. The blank area circled by the green line indicates that the bacterial inhibition rate was over 95%. All data are presented as the mean values of three biological replicates.

$\beta$-lactamase when used alone but exhibited excellent $\beta$-lactamase inhibition when used in combination with rifampicin (27). Therefore, the combined antibacterial effect of GE-EE with penicillin G was likely to depend on the interaction of kaempferol glycosides and quercetin with penicillin G. The combination of GE-EE with ceftazidime, erythromycin, levofloxacin, or vancomycin did not show a synergistic inhibitory activity against MRSA strains.

Moreover, GE-EE combined with penicillin G in various combinations tested significantly inhibited biofilm formation by *S. aureus* strains BWSA11, BWSA15, and ATCC 43300, respectively, and most of the combinations synergistically inhibited biofilm formation by *S. aureus* BWSA15 and ATCC 43300 (Fig. 6). The combination of GE-EE and penicillin G did not show a synergistic inhibitory effect on the biofilm of BWSA11, which may be attributed to the extremely sensitive nature of biofilm formation by *S. aureus* BWSA11 to GE-EE at all concentrations tested (Fig. 6). It was further observed that kaempferol-3,7-dirhamnoside (15.63 to 250 $\mu$g/mL) combined with penicillin G (16 to 128 $\mu$g/mL) also significantly or synergistically inhibited biofilm formation by *S. aureus* BWSA11 (Fig. 6). Unlike the combination of kaempferol-3,7-dirhamnoside and

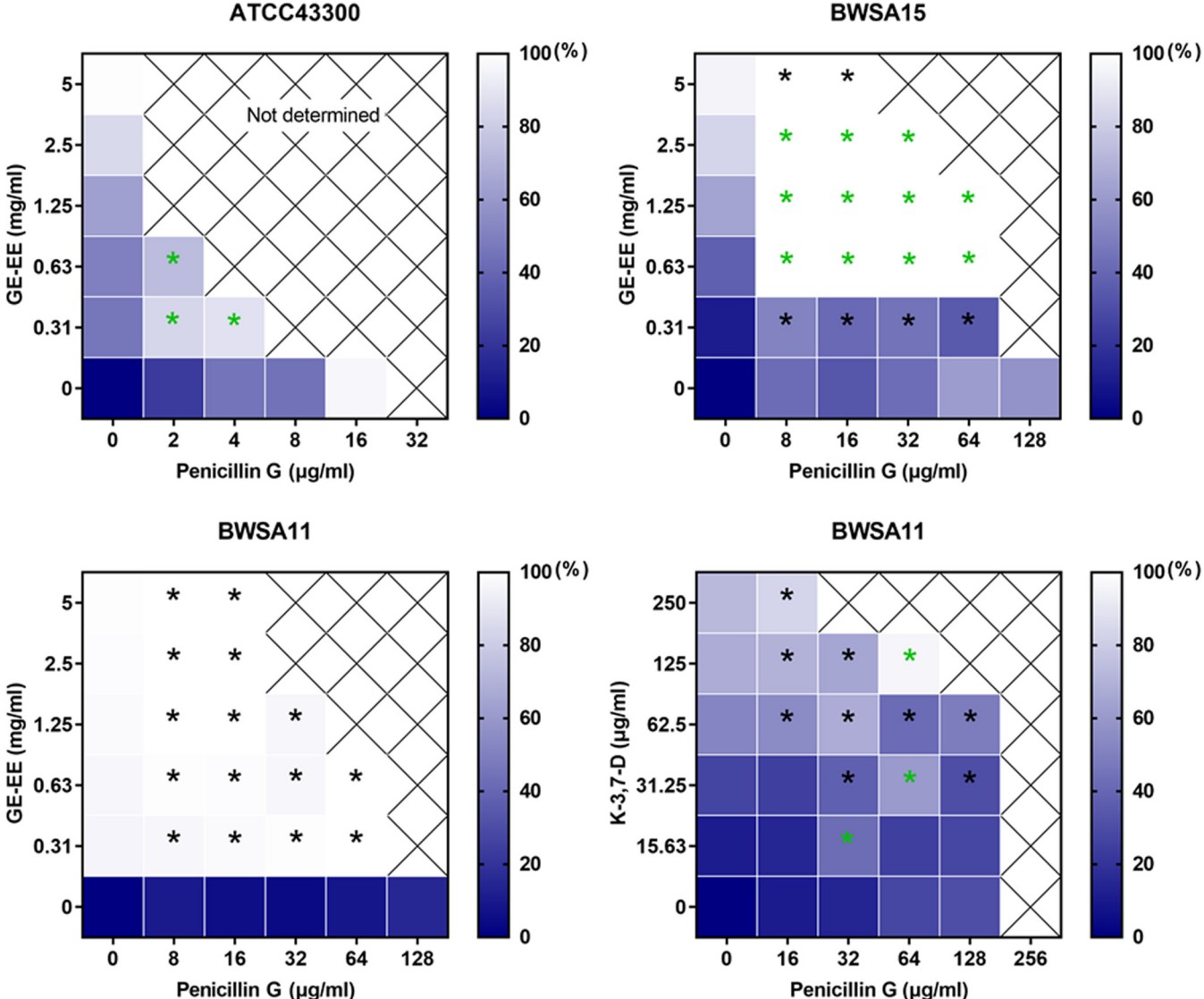

**FIG 6** Heat map of antibiofilm effects of penicillin G in combination with GE-EE and representative components. GE, ear of *C. ambrosioides* L. collected from Guangxi, China; EE, ethanol extracts; K-3,7-D, kaempferol-3,7-dirhamnoside; Not determined, the biofilm formation of the area where the bacterial inhibition rate was over 95% was not calculated. Black asterisks indicate significant decreases at a *P* value of 0.05 compared to the control. Green asterisks indicate significant decreases at a *P* value of 0.05 compared to the control and either of the two parallel single-agent groups. All data are presented as the mean values from three biological replicates.

penicillin G, the combination of quercetin and penicillin G at certain concentrations instead induced biofilm formation by *S. aureus* BWSA11 (Fig. S3). Therefore, the synergistic inhibitory effect of GE-EE and penicillin G on the biofilm formation of representative strains of *S. aureus* was also unlikely to be the result of the interaction between penicillin G and a single component.

**Global downregulation of penicillin resistance and biofilm formation by *S. aureus* in response to GE-EE was with *agr*, *sarA*, and *sigB* rather than *luxS*.** A total of 17 genes functionally contributing to penicillin resistance and/or biofilm formation (Table S2) were selected to determine the transcriptional response of *S. aureus*. As shown in Fig. 7, these contributors are transcriptionally interconnected to *agr* (28–31), *luxS* (32–34), *sarA* (35–37), and *sigB* (38–40). Specifically, for *agr*-dependent regulation of PSM, the dramatic effect of *agr* on *psmα* expression is mediated by the direct binding of the AgrA response regulator, which occurs independently of RNAIII, the small regulatory RNA (41). It was found that the expression of all these contributors and their potential regulators was downregulated (Fig. 8). Although *agr*, *sarA*, and *sigB* influence each

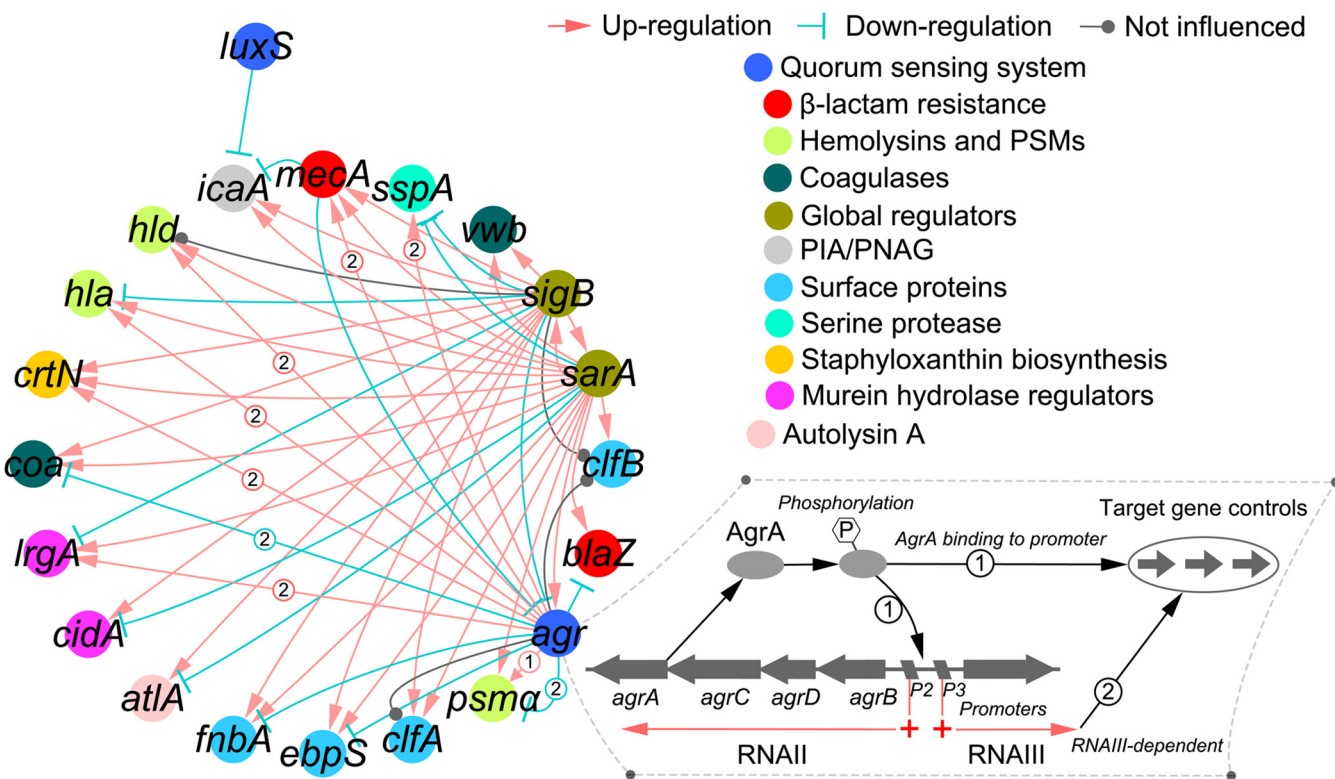

**FIG 7** Network of transcriptional regulation of genes involved in *S. aureus* penicillin resistance and biofilm formation.

other (42, 43), from the perspective of an exclusive positive regulatory relationship, their regulation of *sspA*, *blaZ*, *atlA*, and *cidA* is relatively independent (Fig. 7). Molecular docking can predict the binding ability of molecules by studying the intermolecular interactions. The smaller the molecular docking affinity constant is, the more stable is the binding of the ligand to the receptor protein, indicating that the drug component has a

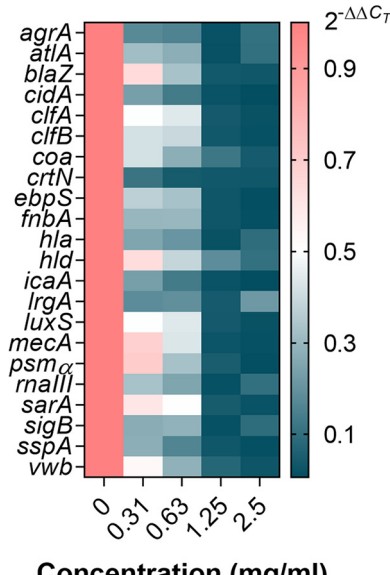

**FIG 8** Heat map of gene expression of *S. aureus* ATCC 43300 in response to GE-EE. GE, ear of *C. ambrosioides* L. collected from Guangxi, China; EE, ethanol extracts. All data are presented as the mean values from three biological replicates.

**TABLE 3** Contributions of genes to penicillin resistance and biofilm formation and potential transcriptional regulation in *S. aureus* ATCC 43300 in response to GE-EE

| | Contribution to[a]: | | Regulation associated with indicated global regulatory gene (% of positively regulated contributors)[b] | | | |
|---|---|---|---|---|---|---|
| Gene | Penicillin resistance | Biofilm formation | *agr* (25) | *sarA* (83) | *sigB* (92) | *luxS* |
| *atlA* | − | + | | N | P | |
| *blaZ* | + | | N | P | | |
| *cidA* | − | + | | N | P | |
| *clfA* | | + | N | P | P | |
| *clfB* | | + | N | P | P | |
| *coa* | | + | N | P | P | |
| *crtN* | | + | P | P | P | |
| *ebpS* | | + | N | P | P | |
| *fnbA* | | + | N | P | P | |
| *icaA* | | + | | P | P | N |
| *hla* | | + | P | P | N | |
| *hld* | | − | P | P | N | |
| *lrgA* | + | − | P | P | N | |
| *mecA* | + | + | P | P | P | |
| *psmα* | | − | P | P | N | |
| *sspA* | | − | P | N | N | |
| *vwb* | | + | N | P | P | |

[a]−, negative contributor; +, positive contributor.
[b]The values in parentheses are the percentages of positively regulated contributors to biofilm formation associated with each regulatory gene. N, negative regulation; P, positive regulation.

strong binding force to the key protein. RsbU phosphatase activation is critical for *sigB*-dependent transcription (44). Judging from the higher docking scores (Table S3), the flavonoids represented by kaempferol glycosides had strong binding ability with the RsbU, AgrA, and SarA proteins. Moreover, given the negative regulatory relationship between *luxS* and *icaA* (Fig. 7), the downregulation of *icaA* is unlikely to be the result of reduced transcription of *luxS*. In addition to *icaA*, the association of *luxS* with other genes is rarely reported. Therefore, the global downregulation of all genes related to penicillin resistance and biofilm formation observed in this study is likely the result of the interaction of kaempferol glycosides and quercetin with *agr*, *sarA*, and *sigB*.

**GE-EE-induced downregulation of penicillinase was mainly responsible for the increased susceptibility of MRSA to penicillin G.** As summarized in Table 3, penicillin resistance is associated not only with the production of penicillinase (*blaZ*), but also with the presence of penicillin-binding protein 2a (PBP2a) (*mecA*), a transpeptidase enzyme that presents very low beta-lactam affinity. Furthermore, murein hydrolysis caused by upregulation of CidA (*cidA*) (45) or the positive regulator of autolysis Atl (*atlA*) or downregulation of the negative regulator LrgA (*lrgA*) (46) may also be associated with penicillin resistance. In this study, contributors including *blaZ*, *mecA*, *cidA*, and *atlA* were found to be transcriptionally downregulated in *S. aureus* ATCC 43300 in the presence of GE-EE (0.31 to 2.5 mg/mL) (Fig. 8). Inactivation of LuxS/AI-2 has also been reported to result in decreased autolysis and decreased susceptibility to cell wall synthesis inhibitor antibiotics like penicillin, oxacillin, vancomycin, and teicoplanin (34, 47). Although *mecA* expression was significantly downregulated, the bacteriostatic activity of ceftazidime, a *β*-lactam antibiotic, and vancomycin against *S. aureus* ATCC 43300 was contrarily found to be reduced (data not shown). This observation was consistent with decreased expression of *luxS*, *cidA*, and *atlA*, indicating decreased cell wall hydrolytic activity of *S. aureus* ATCC 43300 in response to GE-EE. Therefore, the increased susceptibility of *S. aureus* ATCC 43300 to penicillin G was mainly due to reduced production of penicillinase in response to GE-EE. Given that the synchronous expression (Fig. 9) of *agr* (*agrA* and *rnaIII*) and *blaZ* contradicts their negative transcriptional relationship (Fig. 7), it is likely that *sarA* was primarily responsible for the downregulation of penicillinase.

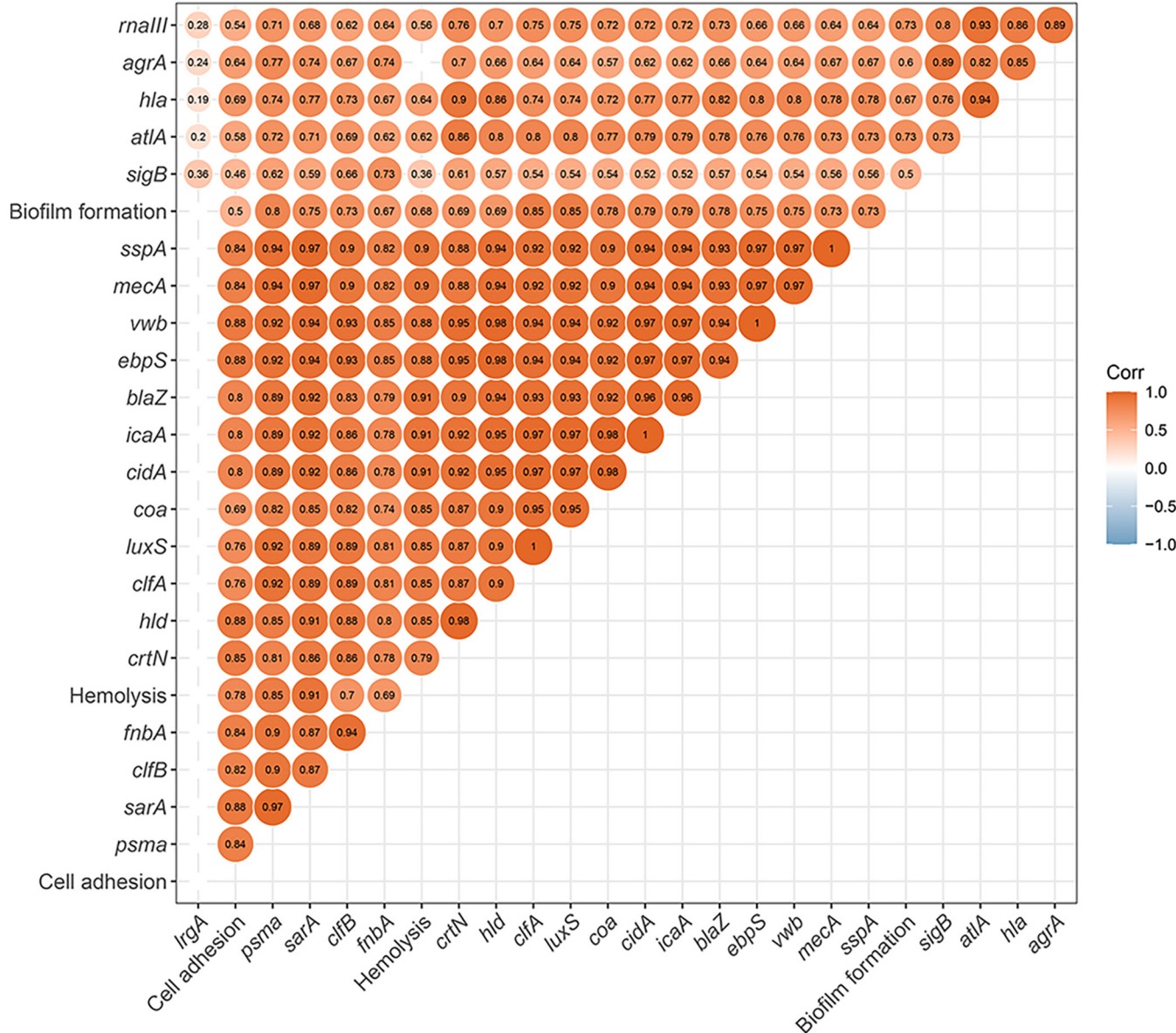

**FIG 9** Matrix graph of correlations (Corr) among gene expression, biofilm formation, hemolysis, and cell adhesion in *S. aureus* ATCC 43300 in response to GE-EE. GE, ear of *C. ambrosioides* L. collected from Guangxi, China; EE, ethanol extracts. Significant correlation was calculated at the level of *P* < 0.05.

**GE-EE targeting *sarA* and *sigB* downregulated multiple virulence factors involved in the initial and mature stages of biofilm formation.** Among contributors observed in this study, *atlA, cidA, clfA, clfB, coa, crtN, ebpS, fnbA, hla, icaA, mecA,* and *vwb* can contribute to *S. aureus* biofilm formation by directly or indirectly mediating surface adhesion or intercellular aggregation or protecting bacterial cells from immune clearance (Table S2), which mechanistically involves various aspects of biofilm formation. The multitargeted antibiofilm property of GE-EE was also corroborated by significant reductions in cell adhesion (Fig. 9 and 10), erythrocyte lysis (Fig. 9 and 11 and Fig. S4), and PIA production (Fig. 9 and Fig. S5) by *S. aureus* ATCC 43300 and clinical representatives. Certain flavonoids can effectively inhibit toxic expressions like α-hemolysin (48). The downregulation of *hld, psmα,* and *sspA* suggests that the inhibition of *S. aureus* ATCC 43300 biofilm by GE-EE was independent of dissociation activity. Furthermore, *agr* (30, 36) was found to be associated with positive downregulation of *crtN, hla,* and *mecA,* accounting for 25% of biofilm positive contributors, while *sigB* and *sarA* were associated

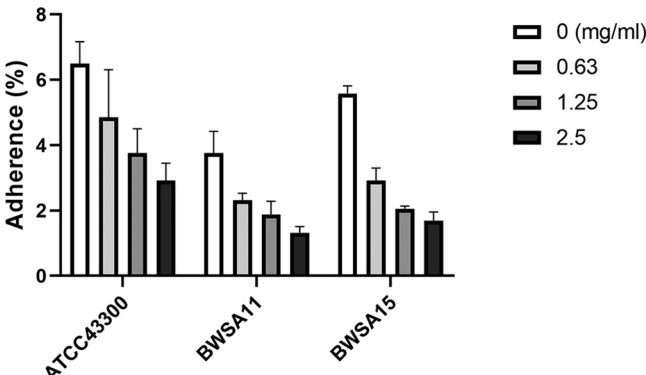

**FIG 10** Cell adhesion activities of *S. aureus* strains in the presence of GE-EE. GE, ear of *C. ambrosioides* L. collected from Guangxi, China; EE, ethanol extracts. All data are presented as the mean values from three biological replicates. Error bars show SD.

with 92% and 83% of biofilm positive contributors, respectively (Table 3). Exceptionally, *mecA* (49) can in turn inhibit the activity of *agr* (Fig. 7), thereby indirectly affecting the expression of related contributors. But judging from the synchronized expression of *mecA* and *agr* (Fig. 9), the downregulation of *mecA* was likely to be a concomitant phenomenon rather than a cause of the downregulation of biofilm expression. It has been reported that extracellular DNA released from Atl-dependent autolysis is mainly responsible for the early stages of MRSA biofilm formation (50), whereas the FnBPs promote subsequent intercellular accumulation and biofilm maturation (51). This observation is consistent with the downregulation of *atlA* and *fnbA* and accompanying biofilm attenuation in *S. aureus* ATCC 43300 in response to GE-EE. Taking into account the regulatory relationship among *atlA*, *fnbA*, *sarA*, and *sigB*, the downregulation of *atlA* is most likely due to reduced expression of *sigB*, while the downregulation of *fnbA* is likely to be related to both *sigB* and *sarA*.

In conclusion, the synergistic antibacterial and inhibitory biofilm activity of *C. ambrosioides* L. alcohol extract combined with penicillin G against MRSA was closely related to the interaction between the main components of kaempferoside glycosides and quercetin. In mechanism, the increased sensitivity of MRSA to penicillin G was mainly related to the downregulation of penicillinase expression, with SarA as a potential drug target, while the antibiofilm activity was mainly related to downregulation of various virulence factors involved in the initial and mature stages of biofilm development, with SarA and/or $\sigma$B as a potential drug target. This study provides a theoretical basis for further exploration of the medicinal activity of kaempferol rhamnosides and quercetin and their application in combination with the classic old drug penicillin G against MRSA biofilm infection.

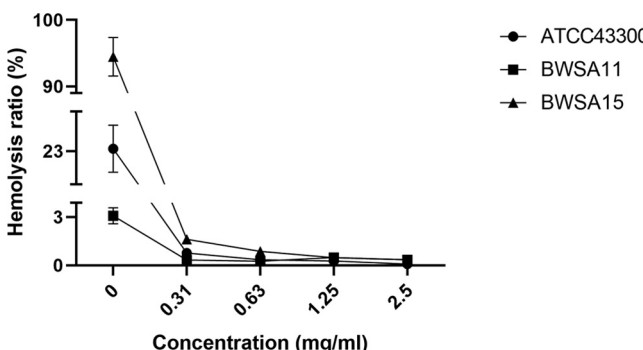

**FIG 11** Hemolytic activities of *S. aureus* strains in the presence of GE-EE. GE, ear of *C. ambrosioides* L. collected from Guangxi, China; EE, ethanol extracts. All data are presented as the mean values from three biological replicates. Error bars show SD.

## MATERIALS AND METHODS

**Plant materials and strains.** The *C. ambrosioides* L. plants were collected from Qinzhou prefecture in Guangxi, China. Fresh plant materials were separated into three parts, including root (GR), stem (GS), and ear (GE) with leaves and seeds, dried in the shade after washing, crushed, and stored at −20°C prior to use. Three MRSA strains, including the standard strain *S. aureus* ATCC 43300 and two clinical strains, BWSA11 and BWSA15, isolated from burn wounds, were used in this study.

**Preparation of CAEs.** *C. ambrosioides* L. extracts (CAEs) were prepared by the immersion extraction method, using water and ethanol as the solvents. For water extracts (WE), the dried and crushed plant samples were added to pure water at a solid-to-liquid ratio of 1:10 (g/vol), extracted for 48 h, filtered by centrifugation, concentrated under reduced pressure, and freeze-dried to obtain GE-WE, GS-WE, and GR-WE powders, respectively. For ethanol extracts (EE), extracts GE-EE, GS-EE, and GR-EE were prepared by adding 50% ethanol to different plant parts at a ratio of 3:40 (g/vol) with ultrasonic extraction for 30 min two times. The extracts showing the highest antibiofilm formation activity were further extracted using fractional extraction with petroleum ether (PE), ethyl acetate (EA), and *n*-butanol (NB), respectively. All extracts, including the residual aqueous layer (RA), were subjected to filtration, reduced-pressure concentration, and freeze-drying. The powders obtained were dissolved in sterile water to achieve a high-concentration reserve solution (40 mg/mL) prior to use.

**Antibacterial activity assays.** The antibacterial activities were determined by the disk diffusion or microdilution method according to CLSI guidelines (52). In the disk diffusion test, bacterial lawns of *S. aureus* ATCC 43300 were prepared by evenly spreading 50 $\mu$L of bacterial culture (approximately $1.5 \times 10^8$ CFU/mL) on Trypticase soy agar (TSA) with sterile cotton swabs and then placing sterile filter paper discs (4 mm in diameter) impregnated with CAEs (200 $\mu$g/disc) on the agar surface. The inhibition zone diameters were determined after incubation for 20 h at 37°C. Kanamycin was used as a positive control.

For the microdilution test, serial 2-fold dilutions were made in concentrations ranging from 5 mg/mL to 0.3125 mg/mL for CAEs and from 128 $\mu$g/mL to 4 $\mu$g/mL for antibiotics in sterile 96-well microplates. The initial populations of bacterial cells were approximately $5 \times 10^5$ CFU/mL. The MICs for *S. aureus* strains ATCC 43300, BWSA11, and BWSA15 were determined as the lowest concentrations of CAEs or antibiotics at which bacterial growth was reduced by more than 95% after incubation at 37°C for 20 h.

**Biofilm formation assays.** The biofilm formation assays were carried out on 96-well plates according to a previously described method (53) with slight modifications. Briefly, bacterial cells were inoculated into a 200-$\mu$L Trypticase soy broth (TSB) culture system with a final population of approximately $5 \times 10^5$ CFU/mL in each well, in the absence or presence of CAEs. After incubation at 37°C for 24 h, the optical density (OD) values of planktonic cells were measured at 600 nm ($OD_{planktonic}$). The biofilm remaining in each well was stained with a 0.1% crystal violet (CV) (Sigma) solution at 37°C for 8 min. The stained biofilm cells were destained with 95% ethanol and measured at 600 nm ($CV_{biofilm}$). The wells without inoculation were used as the negative controls ($CV_{control}$ and $OD_{control}$) to reduce the background signals. The ability to form a biofilm was expressed as the biofilm formation index (BFI), as follows: $[BFI = (CV_{biofilm} - CV_{control})/(OD_{planktonic} - OD_{control})]$.

**LC-MS analysis.** The extract showing the highest antibiofilm formation activity was subjected to liquid chromatography-mass spectrometry (LC-MS) analysis. Samples were prepared by being dissolved in methanol and filtrated through 0.22-$\mu$m membrane filters prior to LC-MS analysis. For LC analysis, an Agilent 1200 high-performance liquid chromatography instrument (Agilent Technologies, USA) equipped with a diode array detector and a $C_{18}$ column (460 by 25 mm, 5-$\mu$m particle size) was used. The injection volume was 5 $\mu$L. The temperature of the column oven was 25°C. The detection condition was full-wavelength (200 to 600 nm) scanning. Mobile phase A was 0.1% acetic acid aqueous solution, and mobile phase B was acetonitrile. The gradient program, using a flow rate of 1.0 mL/min, was as follows: 0 to 50 min, from 8% B to 31% B; 50 to 55 min, from 31% B to 100% B; 55 to 60 min, 100% B; 60 to 63 min, from 100% B to 8% B. The conditions for mass spectrometry were set as follows. The atomization pressure of the electrospray ionization (ESI) ion source was 35 lb/in². The nitrogen flow rate was 12 L/min, and the temperature was 325°C. Scanning was performed in an *m/z* range of 100 to 1,500 atomic mass units (amu), with an ionization voltage of 4,000 V and a fragmentation voltage of 250 V in the cation mode and an ionization voltage of 3,500 V and a fragmentation voltage of 175 V in the anion mode.

**Antimicrobial checkerboard assays.** *S. aureus* strains ATCC 43300, BWSA11, and BWSA15 were chosen to investigate the antibacterial activities of CAEs and their representative compounds in combination with ceftazidime, erythromycin, levofloxacin, penicillin G, and vancomycin. A broth microdilution checkerboard procedure (54) with slight modification was used to determine the inhibitory effect of drug combinations on the growth of bacterial cells. Each drug was 2-fold diluted with TSB at six concentrations to create a 6-by-6 matrix in a 96-well plate. The concentrations of antibiotics for which MIC values could not be obtained previously due to antibiotic resistance and detection limitations were set between 4 $\mu$g/mL and 128 $\mu$g/mL. The concentrations of antibiotics for which MIC values had been previously obtained were set between 1/8× MIC and 2× MIC. The interactions between CAEs and antibiotics were determined based on the calculated fractional inhibitory concentration index (FICI) (55). The combined antimicrobial effect was interpreted as synergy (FICI of ≤0.5), additivity (FICI of >0.5 and <1), indifference (FICI of ≥1 and <4), or antagonism (FICI of ≥4). The FICI was confirmed based on three independent replicates of the test.

**Antibiofilm checkerboard assays.** Strains with a FIC of ≤0.5 in response to different drug combinations were further tested to observe the combined effects of drugs on the biofilm formation. Assays were carried out in a 6-by-6 matrix on the basis of the antimicrobial checkerboard assay and the biofilm formation assay described above. The biofilms formed under conditions without exposure to drugs were used as the positive controls. The percentage of inhibition was used to evaluate the combined

drug effects on the biofilm formation compared to that of the positive control. A synergistic inhibitory effect on the biofilm formation was considered when the relative biofilm formation index (RBFI) value of the combined drug group was significantly decreased compared to both the positive control and the corresponding single drug treatments.

**Cell adhesion assays.** The A549 human alveolar epithelial cell line, routinely cultured in Dulbecco's modified Eagle's medium (DMEM; Solarbio), was used to evaluate the cell adhesion ability of bacterial strains according to a method reported by Tang et al. (56), with some modifications. Briefly, a monolayer of approximately $5 \times 10^4$ A549 cells was prepared in a 24-well plate prior to use. A bacterial suspension with a population of approximately $5 \times 10^5$ CFU in 1 mL of DMEM with or without CAE was aseptically transferred to each well at 37°C. The A549 cells were washed three times with phosphate-buffered saline (PBS) after 1 h of incubation and lysed with 500 $\mu$L of solubilizing solution (400 $\mu$L 0.5% Triton X-100 and 100 $\mu$L 0.25% trypsin-EDTA solution). Dilutions of lysates were spread on TSA, and the bacterial numbers were counted after incubation at 37°C for 24 h.

**Hemolytic assays.** Defibrinated blood was prepared from a New Zealand rabbit prior to use. TSB medium with a bacterial inoculation of approximately $1 \times 10^6$ CFU/mL in the absence or presence of CAE was incubated at 37°C for 20 h. The culture supernatant was prepared by centrifugation ($5,000 \times g$) at 4°C for 5 min, followed by filtration using a Millipore membrane filter (0.22 $\mu$m). A mixture containing 200 $\mu$L bacterial supernatant, 775 $\mu$L $1\times$ PBS, and 25 $\mu$L defibrinated rabbit blood was incubated at 37°C for 30 min. The supernatant of the mixture was obtained by centrifugation at $5,000 \times g$ for 1 min, and the absorbance was measured at 543 nm ($A_{treatment}$). A mixture without bacterial inoculation was used as a negative control to reduce the background signal ($A_{n-control}$). A positive control ($A_{p-control}$) was obtained by mixing 975 $\mu$L of distilled water and 25 $\mu$L of defibrinated rabbit blood. The hemolytic activity of the tested strain was expressed as the hemolysis ratio as follows: hemolysis ratio (%) = ($A_{treatment} - A_{n-control}$)/ $A_{p-control} \times 100$.

**PIA production assay.** The production of polysaccharide intercellular adhesin (PIA) was evaluated by using the Congo red agar method. Briefly, bacterial cells, after spot inoculation onto TSA agar with Congo red (50 $\mu$g/mL) in wells of 24-well plates, were incubated at 37°C for 24 h. The formation of black and brown lawns indicated that bacteria produced PIA, and red lawns indicated that bacteria did not produce PIA.

**RT-PCR assays.** The transcriptional profile of *S. aureus* ATCC 43300 in response to CAEs was assessed in this study. *S. aureus* ATCC 43300 was chosen to evaluate the transcriptional changes of genes, including *icaA*, *agrA*, *rnaIII*, *sarA*, *luxS*, *fnbA*, *fnbB*, *clfA*, *clfB*, *ebpS*, *cidA*, *lrgA*, *sspA*, *psm*, *atlA*, *hld*, *vwb*, and *coa*, in response to CAEs. Briefly, the bacterial cells were cultivated in TSB in the absence or presence of GE-EE (2.5 mg/mL) at 37°C for 24 h. After incubation, the bacterial cells were collected by centrifugation at $5,000 \times g$ for 5 min and then resuspended with Tris-EDTA (TE) buffer. Total RNA was extracted using TRIzol (Solarbio Bio, Inc., Beijing, China). The cDNA was synthesized using the TianGen FastKing genomic DNA (gDNA)-dispelling reverse transcription (RT) supermix kit according to the manufacturers' protocol. The PCR amplification was performed in a 20-$\mu$L reaction mixture containing 10 $\mu$L of $2\times$ QuantiTect SYBR green PCR master mix (Qiagen), 2 $\mu$L of cDNA, and 10 pmol/$\mu$L each primer (Table S1), using the ABI 7500 real-time PCR system (Applied Biosystems, Foster City, CA). The 16S rRNA gene was used as the reference gene, and the comparative threshold cycle ($\Delta C_T$) method was used to analyze the relative expression levels of the targeted genes.

**Molecular docking simulation.** Molecular docking (57) was performed to assess the affinity of interaction between compounds identified from CAEs and the IcaA, AgrA, RNAIII, SarA, LuxS, FnbA, FnbB, ClfA, ClfB, EbpS, CidA, LrgA, SspA, Psm, AtlA, Hld, Vwb, and Coa proteins. Crystallographic structures of the selected proteins were downloaded from RCSB Protein Data Bank (https://www.rcsb.org/) with their native ligands. The ligand small molecules obtained by LC-MS analysis were all from the PubChem database (https://pubchem.ncbi.nlm.nih.gov/), and the conformation was optimized by using Chem3D 20.0 for molecular mechanics (MM2). The ligand small-molecule materials were imported into AutoDock Tools 1.5.6 for hydrogenation, charge calculation, root detection, and rotatable bond searching and definition. The three-dimensional (3-D) structure of the protein downloaded from the RCSB was used as the docking protein, and it was stored as the receptor in PDBQT format after hydrogenation, Gasteiger charge calculation, and nonpolar hydrogen combination. Molecular docking was carried out with the help of Autodock Vina 1.1.2 software. The conformation with the highest docking affinity (kcal/mol) was selected as the final docking conformation for semiflexible docking, and PyMOL software was used for visual mapping.

**Statistical analysis.** The data collected were analyzed using SPSS 25 and GraphPad Prism 8. All data are presented as the mean values $\pm$ standard deviations (SD). One-way analysis of variance (ANOVA) was used to determine the differences between the test groups.

## SUPPLEMENTAL MATERIAL

Supplemental material is available online only.

**SUPPLEMENTAL FILE 1**, PDF file, 0.9 MB.

## ACKNOWLEDGMENTS

X.H. analyzed the data and drafted the manuscript. W.Z. and Q.C. contributed to almost all data collection. Y.L. contributed to the cell adhesion assay. G.B., T.L., J.B., and C.C. participated in the analysis of data. C.Y. contributed to the preparation of

*C. ambrosioides* L. extracts. Y.Y., J.X., Z.R., Y.J., and F.L. analyzed the data and revised the manuscript. All authors read and approved the final manuscript.

This study was supported by the National Nature Science Foundation of China (grant no. 82072297), the Open Project Program of Jiangsu Key Laboratory of Zoonosis (grants no. R1908 and R2109), the Six Talent Peaks Project in Jiangsu Province (grant no. 2019-YY-065), and the High Level Talent Support Project of Yangzhou University.

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
