## [Reviewer comments · Microbiology Spectrum]

Microbiology Spectrum

Global down-regulation of penicillin resistance and biofilm formation by MRSA is associated with the interaction between kaempferol rhamnosides and quercetin

Xinlong He, Wenwen Zhang, Qingchao Cao, Yinyue Li, Guangyu Bao, Tao Lin, Jiaojiao Bao, Caiwang Chang, Changshui Yang, Yi Yin, Jiahui Xu, Zhenyu Ren, Yingshan Jin, and Feng Lu

Corresponding Author(s): Feng Lu, Yangzhou University

Review Timeline:

Submission Date:	July 20, 2022
Editorial Decision:	September 13, 2022
Revision Received:	September 21, 2022
Accepted:	October 24, 2022

Editor: Cezar Khursigara

Reviewer(s): The reviewers have opted to remain anonymous.

Transaction Report:

DOI: <https://doi.org/10.1128/spectrum.02782-22>

September 13, 2022

Dr. Feng Lu
Yangzhou University
Department of Pathogenic Biology, School of Medicine
Yangzhou
China

Re: Spectrum02782-22 (Global down-regulation of penicillin resistance and biofilm formation by MRSA is associated with the interaction between kaempferol rhamnosides and quercetin)

Dear Dr. Feng Lu:

Two experts and I have reviewed your manuscript and agree that although the findings and quality are suitable for Spectrum, modifications are required before publication. Therefore, please address all the reviewer's comments when submitting a revised version of your manuscript.

Link Not Available

Sincerely,

Cezar Khursigara

Journals Department
Reviewer comments:

Reviewer #1 (Comments for the Author):

This study is well designed and written based on the results obtained. There are few minor comments.

1. In Table 1, add the concentration of each compound with relative proportions.
2. State the reason why the penicillin was used rather than other generations of beta-lactams or other classes of antibiotics.
3. Tables 2 and 3 should be combined.
4. Use "anti-biofilm activity" instead of biofilm inhibitory activity throughout the manuscript.

Reviewer #3 (Comments for the Author):

more reliable result could be obtained if clinical samples increased and diverted
In methodology there is few grammar mistakes and better to avoid repetition.
regulation of gene not discussed in details (the Method)

Staff Comments:

Preparing Revision Guidelines

Please return the manuscript within 60 days; if you cannot complete the modification within this time period, please contact me. If you do not wish to modify the manuscript and prefer to submit it to another journal, please notify me of your decision immediately so that the manuscript may be formally withdrawn from consideration by Microbiology Spectrum.

Global down-regulation of penicillin resistance and biofilm formation by MRSA is associated with the interaction between kaempferol rhamnosides and quercetin

Xinlong He^{1, 3, 5, 6, 7, †}, Wenwen Zhang^{1, †}, Qingchao Cao^{2, †}, Yinyue Li¹, Guangyu Bao³, Tao Lin³, Jiaojiao Bao³, Caiwang Chang³, Changshui Yang⁴, Yi Yin¹, Jiahui Xu¹, Zhenyu Ren¹, Yingshan Jin^{2, 5, *}, Feng Lu^{1, 3, 5, 6, 7, *}

¹ Department of Pathogenic Biology, School of Medicine, Yangzhou University, Yangzhou 225009, P. R. China.

² College of Bioscience and Biotechnology, Yangzhou University, Yangzhou 225009, P. R. China.

³ Affiliated Hospital of Yangzhou University, Yangzhou 225000, P. R. China.

⁴ Department of Pharmacy, School of Medicine, Yangzhou University, Yangzhou 225009, P. R. China.

⁵ Jiangsu Key Laboratory of Zoonosis, Yangzhou University, Yangzhou 225009, P. R. China.

⁶ Jiangsu Key Laboratory of Experimental & Translational Non-coding RNA Research, School of Medicine, Yangzhou University, Yangzhou, 225009, P. R. China.

⁷ Jiangsu Co-Innovation Center for the Prevention and Control of Important Animal Infectious Diseases and Zoonoses, College of Veterinary Medicine, Yangzhou University, Yangzhou 225009, P. R. China.

* Corresponding author. Tel.: +86 136 4525 1950; E-mail: lufeng@yzu.edu.cn (F.L.) and ysjin@yzu.edu.cn (Y.J.)

† These authors contributed equally to this work.

Running title: Transcriptional down-regulation of MRSA

ABSTRACT The rapid development of MRSA drug resistance and the formation of

biofilms seriously challenge the clinical application of classic antibiotics. The extracts of the traditional herb *Chenopodium ambrosioides* L. were found to have strong biofilm inhibitory activity against MRSA, but its mechanism of action remains poorly understood. This study was designed to investigate the antibacterial and anti-biofilm activities of flavonoids identified from *C. ambrosioides* L. in combination with classic antibiotics including penicillin G against MRSA. LC-MS was used to analyze the nonvolatile chemical compositions. Reverse transcription-PCR was used to investigate potential multi-targets of flavonoids based on global transcriptional responses of virulence and antibiotic resistance. A synergistic antibacterial and biofilm-inhibiting activity of the alcohol extract of *C. ambrosioides* L. in combination with penicillin G was observed against MRSA, which proved to be closely related to the interaction of the main components of kaempferol rhamnosides with quercetin. In mechanism, the increased sensitivity of MRSA to penicillin G was shown to be related to the down-regulation of penicillinase with SarA as a potential drug target, while the biofilm inhibitory activity is mainly related to down regulation of various virulence factors involved in the initial and mature stages of biofilm development with SarA and/or σ B as drug targets. This study provides a theoretical basis for further exploration of the medicinal activity of kaempferol rhamnosides and quercetin and its application in combination with penicillin G against MRSA biofilm infection.

IMPORTANCE In this study, the synergistic antibacterial and anti-biofilm effects of the traditional herb *C. ambrosioides* L. and the classic antibiotic penicillin G on MRSA provide a potential strategy to deal with the rapid development of MRSA antibiotic resistance. This study also provides a theoretical basis for further optimizing the combined effect of kaempferol rhamnosides, quercetin, and penicillin G, and

exploring the anti-MRSA biofilm infection research with SarA and σ B as drug targets.

KEYWORDS *Staphylococcus aureus*, MRSA, biofilm formation, antibiotic resistance, global regulation, *Chenopodium ambrosioides* L., kaempferol rhamnosides, quercetin

INTRODUCTION

Staphylococcus aureus, the most common pathogen in human infection, can cause local purulent infection, pneumonia, pseudomembranous colitis, pericarditis, and even sepsis(1). Soon after penicillin was introduced for therapeutic use in 1940s, an increased incidence of penicillin resistance was reported in *S. aureus* strains (2). Methicillin, a semi-synthetic penicillin, is resistant to penicillinase, and unfortunately, methicillin-resistant *S. aureus* (MRSA) emerged shortly after its clinical application (3). Today, MRSA is the leading cause of nosocomial infections (4). The prevalence of MRSA infection has resulted in the marginalization of the clinical use of traditional classic antibiotics including penicillin. On the other hand, the current development of new drugs is seriously lagging behind the rapid development of drug resistance, and the emergence of completely drug-resistant superbugs is bound to put patients in an incurable dilemma. Repurposing of old drugs with the idea of reducing toxicity is now considered as a potentially effective strategy to control MRSA infection (5, 6).

Biofilms are microbial communities embedded in an extracellular matrix (ECM) composed of lipids, proteins, polysaccharides, and DNA that can form on medical implants or tissue surfaces to protect pathogenic bacteria from immune clearance and antibiotic killing, and are therefore implicated in most chronic infections (7, 8). Biofilms are formed in multi-stage process including initial attachment, accumulation, maturation and dispersion, and various virulence factors are involved in the progress along the stages of *S. aureus* biofilm formation (9). For example, surface protein

adhesins, typically known as MSCRAMMs (microbial surface components recognising adhesive matrix molecules), including fibrinogen-binding proteins (FnBPs), fibrinogen-binding clumping factors (Clfs), elastin-binding protein (EbpS), and autolysin Atl, confer the ability for *S. aureus* to adhere to the host matrix(10). Polysaccharide intercellular adhesin (PIA), also referred as poly-N-acetylglucosamine (PNAG) contributes to biofilm accumulation (11). Extracellular DNA (eDNA) released by bacterial autolysis can be an important component of the biofilm matrix (12). Proteases and phenol-soluble modulins (PSM) peptides may act as dispersants of biofilm to aid in bacterial dissemination and the settlement of new biofilms at a distant site (13). Moreover, the coordinated expressions of these virulence factors is modulated in complex networks that include the global regulators accessory gene regulator Agr, the staphylococcal accessory regulator SarA, and the alternative sigma factor B (σ B) (14). Therefore, these virulence factors and their regulators are potential targets for effective prevention and control of *S. aureus* biofilm-associated infections.

Flavonoids are one of a large class of plant medicinal components with antioxidant, anti-inflammatory, anti-allergic, anti-cancer, anti-viral and anti-fungal properties (15). Recent studies have shown that, in addition to antibacterial effects, some flavonoids also inhibit bacterial biofilm formation by affecting bacterial adhesion, motility, and quorum sensing (QS) (16). *Chenopodium ambrosioides* L., an annual or perennial herb, is widely distributed on the planet and derived essential oil usually used in folk medicine as antirheumatic, anti-inflammatory, antipyretic, antihelmintic, antifungal, and anti-ulcer agents (17-21). However, the biological activities and mechanisms of action in terms of antibacterial and anti-biofilms remain poorly understood. In the previous stage, we compared the biofilm inhibitory activities of the products of alcohol extraction (AE) and water immersion extraction

(IE) from root (GR), stem (GS), and ear (GE) of *C. ambrosioides* L. originated from Guangxi, China, and found that GE-AE had the most significant inhibitory effect on the biofilm formation by *S. aureus* ATCC43300. It was preliminarily estimated that flavonoids were the main active components. The purpose of this study was to investigate the antibacterial and anti-biofilm activities of flavonoids identified from *C. ambrosioides* L. in combination with traditional antibiotics including penicillin G against MRSA, and to investigate potential multi-targets of flavonoids based on global transcriptional responses of virulence and antibiotic resistance.

MATERIALS AND METHODS

Plant materials and strains. The *C. ambrosioides* L. plants were collected from Qinzhou prefecture in Guangxi, China. Fresh plant materials were separated into three parts including root (GR), stem (GS), and ear (GE) with leaves and seeds, dried in the shade after wash, then crushed, and stored at -20 °C prior to use. Three MRSA strains including standard strain of *S. aureus* ATCC43300, and two clinical strains of BWSA11 and BWSA15 isolated from burn wounds, were used in this study.

Preparation of *C. ambrosioides* L. extracts (CAEs). CAEs were prepared by using immersion extraction (IE) and alcohol extraction (AE), respectively. For IE, the dried and crushed plant samples were added with pure water at a solid-liquid ratio of 1:10 (g/v), extracted for 48 h. filtered by centrifugation, concentrated under reduced pressure, and freeze-dried to obtain GE-IE, GS-IE and GR-IE powders, respectively. For AE, extracts of GE-AE, GS-AE and GR-AE were prepared by adding 50% ethanol to different plant parts at a ratio of 3:40 (g/v) with ultrasonic extraction for 30 min for two times. The extracts showing the highest anti-biofilm formation activity were further extracted using the fractional extraction with petroleum ether (PE), ethyl

acetate (EA), and n-butanol (NB), respectively. All extracts including the residual aqueous layer (RA) were subjected to filtration, reduced pressure concentration, and freeze-drying. The obtained powders were dissolved in sterile water to achieve a high concentration reserve solution (40 mg/mL) prior to use.

Antibacterial activity assays. The antibacterial activities **was** determined by the disk diffusion **or** microdilution method according to the CLSI guidelines (22). In the disk diffusion test, the bacterial lawn of *S. aureus* ATCC43300 was prepared by evenly spreading 50 μ L of bacterial cultures (approximately 1.5×10^8 CFU/mL) on trypticase soy agar (TSA) with sterile cotton swabs, then the sterile filter paper discs (4 mm in diameter) impregnated with CAEs (200 μ g/disc) were placed on agar surface. The diameter inhibition zone was determined after incubation for 20 h at 37°C. Kanamycin was used as a positive control.

For the microdilution test, serial twofold dilutions were made in concentrations ranging from 5 mg/mL to 0.3125 mg/mL for CAEs, and from 128 μ g/mL to 4 μ g/mL for antibiotics in sterile 96-well microplates. The initial populations of bacterial cells were approximately 5×10^5 CFU/mL. The minimum inhibitory concentration (MIC) of *S. aureus* strains of ATCC43300, BWSA11, and BWSA15 was determined at the lowest concentration of CAEs or antibiotics at which bacterial growth was reduced by more than 95% after incubation at 37 °C for 20 h.

Biofilm formation assays. The biofilm formation assays were carried out on 96-well plates according to a previous method (23) with slight modification. Briefly, bacteria cells were inoculated into a 200 μ L TSB culture system with a final population approximately 5×10^5 CFU/mL in the absence or presence of CAEs in each well. After incubation at 37°C for 24 h, the OD values of planktonic cells were measured at 600 nm ($OD_{\text{planktonic}}$). The biofilm remaining in each well was stained

with a 0.1% crystal violet (CV) (Sigma) solution at 37°C for 8 min. The stained biofilm cells were destained with 95% ethanol, and measured at 600 nm (CV_{biofilm}). The wells without inoculation were used as the negative controls (CV_{control} and OD_{control}) to reduce the backgrounds. The ability to form a biofilm was expressed as biofilm formation index [BFI= $(CV_{\text{biofilm}} - CV_{\text{control}})/(OD_{\text{planktonic}} - OD_{\text{control}})$].

LC-MS analysis. The extract showing the highest anti-biofilm formation activity was subjected to LC-MS analysis. Samples were prepared by dissolving in methanol, and filtrated through 0.22- μm membrane filter prior to LC-MS analysis. For LC analysis, a high-performance liquid chromatography 1200 (Agilent Technologies, USA) equipped with a diode array detector and a C18 column (460 \times 25 mm, 5 μm). The injection volume was 5 μL . The temperature of the column oven was 25 °C. The detection condition was full-wavelength (200 ~ 600 nm) scanning. The mobile phase A was 0.1% acetic acid aqueous solution, and mobile phase B was acetonitrile. The gradient programmed as follows: 0 ~ 50 min, from 8% B to 31% B; 50 ~ 55 min, from 31% B to 100% B; 55 ~ 60 min, 100% B; 60 ~ 63 min, from 100% B to 8% B at a flow rate of 1.0 mL/min. The condition for mass spectrometry was set as follows. The atomization pressure of ESI ion source was 35 psi. The nitrogen flow rate was 12 L/min, and the temperature was 325°C. Scanning was performed in a m/z range of 100 ~ 1500 Amu, in the cation mode with an ionization voltage of 4000 V, and a fragment voltage of 250 V, and in the anion mode with an ionization voltage of 3500V, and a fragmentation voltage of 175V.

Antimicrobial checkerboard assays. *S. aureus* strains of ATCC43300, BWSA11, and BWSA15 were chosen to investigate the antibacterial activities of CAEs and the representative compounds in combination with penicillin G, ceftazidime, erythromycin, levofloxacin, and vancomycin, respectively. A broth microdilution

checkerboard procedure (24) with a slight modification was used to determine the inhibitory effect of drug combinations on the growth of bacterial cells. Each drug was 2-fold diluted with TSB at six concentrations to create a 6×6 matrix in a 96-well plate. The concentrations of antibiotics for which MIC values could not be obtained previously due to antibiotic resistance and detection limitations were set between 4 $\mu\text{g}/\text{mL}$ and 128 $\mu\text{g}/\text{mL}$. The concentrations of antibiotics for which MIC values have been previously obtained were set between $1/8 \times \text{MIC}$ and $2 \times \text{MIC}$. The interactions between CAEs and antibiotics were determined based on the calculated fractional inhibitory concentration index (FICI) (25). The combined antimicrobial effect was interpreted as synergy ($\text{FICI} \leq 0.5$), additivity ($\text{FICI} > 0.5$ and < 1), indifference ($\text{FICI} \geq 1$ and < 4), and antagonism ($\text{FICI} \geq 4$). The FICI was confirmed based on three independent replicates of test.

Anti-biofilm checkerboard assays. Strains with $\text{FIC} \leq 0.5$ in response to different drug combinations were further tested to observe the combined effects of drugs on the biofilm formation. Assays were carried out in a 6×6 matrix on the basis of the antimicrobial checkerboard assay and the biofilm formation assay described above. The biofilms formed under conditions without exposure to drugs were used as the positive control. Inhibition percentage was used to evaluate the combined drug effects on the biofilm formation as compared to positive control. Synergistic inhibitory effect on the biofilm formation was considered when the RBFi value of the combined drug group was significantly decreased compared to both the positive control and the corresponding single drug treatments.

Cell adhesion assays. The A549 human alveolar epithelial cell line routinely cultured in Dulbecco's Modified Eagle Medium (DMEM, Solarbio) was used to evaluate the cell adhesion ability of bacterial strains according to a method reported

by Tang, J., *et al.*(26) with some modifications. Briefly, a monolayer of A549 approximately 5×10^4 cells was prepared in a 24-well plate prior to use. Bacterial suspension with a population approximately 5×10^5 CFU in a 1 mL of DMEM medium with or without containing CAE was aseptically transferred to each well at 37 °C. The A549 cells were washed three times with phosphate-buffered saline (PBS) after 1 h of incubation, and lysed with 500 μ L of solubilizing solution (400 μ L 0.5% Triton X-100 and 100 μ L 0.25% Trypsin-EDTA solution). Dilutions of lysate were spread on TSA, and the bacteria number was counted after incubation at 37°C for 24 h.

Hemolytic assays. A defibrated blood was prepared from a New Zealand rabbit prior to use. A TSB medium with bacteria inoculation approximately 1×10^6 CFU/mL in the absence or presence of CAE was incubated at 37°C for 20 h. The culture supernatant was prepared by centrifugation ($5,000 \times g$) at 4°C for 5min, and followed by filtration using a Millipore membrane filter (0.22 μ m). A mixture containing 200 μ L bacterial supernatant, 775 μ L $1 \times$ PBS, and 25 μ L defibrillated rabbit blood was incubated at 37°C for 30 min. A supernatant of the mixture was obtained by centrifugation at $5,000 \times g$ for 1 min, and the absorbance was measured at 543 nm ($A_{\text{treatment}}$). A mixture without bacterial inoculation was used as negative control to reduce the background ($A_{\text{n-control}}$). A positive control ($A_{\text{p-control}}$) was obtained by mixing 975 μ L of distilled water and 25 μ L of defibrillated rabbit blood. The hemolytic activity of tested strain was expressed as a hemolysis ratio (%) = $(A_{\text{treatment}} - A_{\text{n-control}}) / A_{\text{p-control}} \times 100$.

Polysaccharide intercellular adhesin (PIA) production assay. The production of PIA was evaluated by using Congo Red Agar method. Briefly, bacterial cells after spot inoculation on TSB agar with Congo red (50 μ g/ml) in wells of 24-well plate were

incubated at 37 °C for 24 hours. Black and brown lawns formed indicate that bacteria produced PIA, and red lawns formed indicate bacteria did not produce PIA.

Reverse Transcription-PCR assays. The transcriptional profile of *S. aureus* ATCC43300 in response to CAEs was assessed in this study. *S. aureus* ATCC4 3300 was chosen to evaluate the transcriptional changes of genes including *icaA*, *agrA*, *rnaIII*, *sarA*, *luxS*, *fnbA*, *fnbB*, *clfA*, *clfB*, *ebpS*, *cidA*, *lrgA*, *sspA*, *psm*, *atlA*, *hld*, *vwb*, and *coa* in response to CAEs. Briefly, the bacterial cells were cultivated in TSB in the absence or presence of GE-AE (2.5 mg/mL) at 37°C for 24 h. After incubation, the bacterial cells were collected by centrifugation at 5,000 × g for 5 min, and then re-suspended with TE buffer. Total RNA was extracted using Trizol (Solarbio Bio. Inc., Beijing, China). The cDNA was synthesized using TianGen FastKing gDNA Dispelling RT SuperMix Kit according to the manufacturers' protocol. The PCR amplification was performed using the ABI 7500 real-time PCR system (Applied Biosystems, Foster City, CA) in a 20-μL reaction mixture containing 10 μL of 2 × QuantiTect SYBR green PCR master mix (Qiagen), 2 μL of cDNA, and 10 pmol/μL each primer (Table 1). The 16s rRNA was used as the reference gene, and the comparative threshold cycle (ΔC_T) method was used to analyze the relative expression of the targeted genes.

Molecular docking simulation. Molecular docking was carried out to assess the affinity interaction between compounds identified from CAEs and proteins of IcaA, AgrA, RNAIII, SarA, LuxS, FnbA, FnbB, ClfA, ClfB, EbpS, CidA, LrgA, SspA, Psm, AtlA, Hld, Vwb, and Coa. The small ligand molecules were all from PubChem database, and structurally optimized by molecular mechanics (MM2) in a Chem3D 20.0 system. The target proteins were all from RCSB Protein Data Bank (PDB), and their 3D structures were all subjected to processes of adding hydrogen atoms,

calculating Gasteiger charges, and incorporating non-polar hydrogen, etc, using AutodockTools, prior to docking. The conformation achieved by semi-flexible docking with the highest affinity (kcal /mol) in Autodock Vina 1.1.2 system was selected as the final conformation.

Statistical analysis. Data collected were analyzed using SPSS 25 and GraphPad Prism 8. All data were presented as mean \pm standard deviation (SD). One-way analysis of variance (ANOVA) was used to determine the differences between the test groups.

RESULTS AND DISCUSSION

Quercetin optimized biofilm inhibition by kaempferol glycosides against *S. aureus*. Significant inhibitory activity against biofilm formation was observed for all extracts of *C. ambrosioides* L. (Figure S1), but no significant inhibitory activity against the growth of *S. aureus* ATCC43300 (Figure S2). It was observed that, among the 6 products of GS, GR and GE extracted by IE and AE, GE-AE exhibited relatively high biofilm inhibition activity against *S. aureus* ATCC43300, while among the fractionated extracts of GE-AE, GE-AE-EA showed the highest biofilm inhibition activity (Figure S1). In contrast, the biofilm inhibitory activity of GE-AE was significantly higher ($P < 0.05$) than that observed for GE-AE-EA at concentrations ranging from 0.3125 to 2.5 mg/ml (Figure 1). This could be due to chemical changes in various kaempferol glycoside derivatives including kaempferol-3,7-dirhamnoside, kaempferol-3-O-apigenin-7-O-rhamnoside, kaempferol-3-O-acetylapienin-7-O-rhamnoside, as well as other flavonoids, rutin, quercetin, etc. (Table 1). Kaempferol glycosides are common bioactive components in *Chenopodium* (27). The accumulative proportion of kaempferol glycosides in GE-AE

was up to 67.5%, suggesting that it likely to be the main active component for biofilm inhibition.

Furthermore, GE-AE (> 62.5 µg/ml) was found to effectively inhibit biofilm formation by *S. aureus* BWSA11 (Figure 2), a clinical MRSA strain with robust biofilm-forming ability(28). The biofilm-inhibitory activity of GE-AE was further found to be significantly correlated with kaempferol-3,7-dirhamnoside, a representative kaempferol glycoside, but not with quercetin and rutin (Table 2). This result is in agreement with a previous finding that kaempferol inhibited the primary attachment phase of biofilm formation in *S. aureus* (29). It was further found that quercetin (1.95~7.81 µg/ml) and kaempferol-3,7-dirhamnoside (15.63~125 µg/ml) had a synergistic inhibitory effect on *S. aureus* BWSA11 biofilm (Figure 3), while rutin and kaempferol-3,7-dirhamnoside did not (data not shown). This finding could explain the higher biofilm inhibitory activity (Figure S1), but the lower total content of kaempferol glycosides in GE-AE (Table 1). Sub-inhibitory concentrations of antibiotics can induce the formation of *S. aureus* biofilms(30), consistent with this, quercetin (1.95 µg/ml) and kaempferol-3,7-dirhamnoside (15.63 µg/ml) also induced the formation of *S. aureus* BWSA11 biofilms (Figure 2). Unlike quercetin or kaempferol-3,7-dirhamnoside, GE-AE showed no *S. aureus* biofilm-inducing activity at all concentrations tested. It was found that the biofilm-inhibitory activities of quercetin and kaempferol-3,7-dirhamnoside within a range of concentrations (0.49~15.63 µg/ml) against *S. aureus* BWSA11 exhibited opposite changes (Figure 2) and were inversely correlated (Table 2), which was likely to result in some degree of mutual containment in biofilm inhibitory or inducing activity. This might explain the biofilm-inhibitory properties of GE-AE that were different from single active ingredients. Therefore, an appropriate combination of active ingredients such as

quercetin and kaempferol glycosides may be effective in eliminating the risk of biofilm induction posed by inappropriate application of the single ingredients.

Synergistic antibacterial and anti-biofilm activity of non-monomeric kaempferol glycosides in combination with penicillin G against *S. aureus*. It was observed that, with the exception of quercetin ($>7.81 \mu\text{g/ml}$), neither GE-AE, kaempferol-3,7-dirhamnoside nor rutin significantly inhibited the growth of *S. aureus* BWSA11 (Figure 4). This observation was in agreement with the result that kaempferol had less bacteriostatic activity against *S. aureus*(31). Considering the extremely low content of quercetin in GE-AE, it is not difficult to explain why the antibacterial property of GE-AE was closely related to kaempferol glycosides (Table 3). Limited by the solubility of kaempferol-3,7-dirhamnoside, it is difficult to determine whether there was a synergistic or additive antibacterial effect between quercetin and kaempferol-3,7-dirhamnoside against the growth of *S. aureus* BWSA11 (FIC < 1.5). However, as shown in Figure 5, the presence of kaempferol-3,7-dirhamnoside (125 and 250 $\mu\text{g/ml}$) and quercetin (15.63 $\mu\text{g/ml}$) reduced each other's MIC value against *S. aureus* BWSA11 by 2-fold and at least 8-fold. Therefore, a proper compatibility of kaempferol glycosides and quercetin is beneficial to the mutual promotion of their antibacterial activities. Furthermore, GE-AE in combination with penicillin G was found to synergistically inhibit the growth of BWSA11, BWSA15, and ATCC43300 with FICs of < 0.19 , < 0.5 , and < 0.16 , respectively (Figure 5). Interestingly, neither the combination of quercetin and penicillin G nor the combination of kaempferol-3,7-dirhamnoside and penicillin G appeared to synergistically inhibit the growth of representative strains tested in this study (data not shown). This is in agreement with a previous finding that kaempferol

and quercetin had mild inhibitory effects on β -lactamase when used alone, but exhibited excellent β -lactamase inhibition when used in combination with rifampicin (32). Therefore, the combined antibacterial effect of GE-AE with penicillin G was likely to depend on the interaction of kaempferol glycosides and quercetin with penicillin G. The combination of GE-AE with other antibiotics did not show a synergistic inhibitory activity against MRSA strains.

Moreover, GE-AE combined with penicillin G at various combinations tested significantly inhibited the biofilm formation by *S. aureus* BWSA11, BWSA15, and ATCC43300, respectively, and most of the combinations synergistically inhibited the biofilm formation by *S. aureus* BWSA15 and ATCC43300 (Figure 6). The combination of GE-AE and penicillin G did not show a synergistic inhibitory effect on the biofilm of BWSA11, which may be attributed to the extremely sensitive nature of biofilm formation by *S. aureus* BWSA11 to GE-AE at all concentrations tested (Figure 6). It was further observed that kaempferol-3,7-dirhamnoside (15.63~250 $\mu\text{g/ml}$) combined with penicillin G (16~128 $\mu\text{g/ml}$) also significantly or synergistically inhibited the biofilm formation by *S. aureus* BWSA11 (Figure 6). Unlike the combination of kaempferol-3,7-dirhamnoside and penicillin G, the combination of quercetin and penicillin G at certain concentrations instead induced the biofilm formation by *S. aureus* BWSA11 (Figure S3). Therefore, the synergistic inhibitory effect of GE-AE and penicillin G on the biofilm formation of representative strains of *S. aureus* was also unlikely to be the result of the interaction between penicillin G and a single component.

Global down regulation of penicillin resistance and biofilm formation by *S. aureus* in response to GE-AE was with *agr*, *sarA*, and *sigB* rather than *luxS*. A total of 17 genes functionally contributing to penicillin resistance and/or biofilm

formation (Table S2) were selected to determine the transcriptional response of *S. aureus*. As shown in Figure 7, these contributors are transcriptionally interconnected to *agr* (33-36), *luxS* (37-39), *sarA* (40-42), and *sigB* (43-45). Specifically for *agr*-dependent regulation of PSM, the dramatic effect of *agr* on *psmA* expression is mediated by the direct binding of the AgrA response regulator, which occurs independently of RNAIII, the small regulatory RNA (46). It was found that the expression of all these contributors and their potential regulators was down-regulated (Figure 8). Although *agr*, *sarA*, and *sigB* influence each other (47, 48), from the perspective of exclusive positive regulatory relationship, their regulation of *sspA*, *blaZ*, *atla*, and *cidA* is relatively independent (Figure 7). Molecular docking can predict the binding ability of molecules by studying the intermolecular interactions. The smaller the molecular docking affinity constant is the more stable the ligand binds to the receptor protein, indicating that the drug component has a strong binding force to the key protein. RsbU phosphatase activation is critical for *sigB*-dependent transcription(49). Judging from the higher docking scores (Table S3), the flavonoids represented by kaempferol glycosides and had strong binding ability with RsbU, AgrA, and SarA proteins. Moreover, given the negative regulatory relationship between *luxS* and *icaA* (Figure 7), the down-regulation of *icaA* is unlikely to be the result of reduced transcription of *luxS*. In addition to *icaA*, the association of *luxS* with other genes is rarely reported. Therefore, the global down-regulation of all genes related to penicillin resistance and biofilm formation observed in this study is likely the result of the interaction of kaempferol glycosides and quercetin with *agr*, *sarA*, and *sigB*.

GE-AE-induced down-regulation of penicillinase was main responsible for increased susceptibility of MRSA to penicillin G. As summarized in Table 4,

penicillin resistance is associated not only with the production of penicillinase (*blaZ*), but also with the presence of PBP2a (*mecA*), a transpeptidase enzyme that presents very low beta-lactam affinity. Furthermore, murein hydrolysis caused by up-regulation of CidA (*cidA*)(50), the positive regulator of autolysis Atl (*atlA*), or down-regulation of the negative regulator LrgA (*lrgA*) (51) may also be associated with penicillin resistance. In this study, contributors including *blaZ*, *mecA*, *cidA*, and *atlA* were found to be transcriptionally down-regulated in *S. aureus* ATCC43300 in the presence of GE-AE (0.31~2.5 mg/mL) (Figure 8). Inactivation of LuxS/AI-2 has also been reported to result in decreased autolysis and decreased susceptibility to cell wall synthesis inhibitor antibiotics such as penicillin, oxacillin, vancomycin, and teicoplanin (39, 52). Although *mecA* expression was significantly down-regulated, the bacteriostatic activity of ceftazidime, a β -lactam antibiotic, as well as vancomycin against *S. aureus* ATCC43300 was contrarily found to be reduced (data not shown). This observation was consistent with decreased expression of *luxS*, *cidA* and *atlA*, indicating a decreased cell wall hydrolytic activity of *S. aureus* ATCC43300 in response to GE-AE. Therefore, the increased susceptibility of *S. aureus* ATCC43300 to penicillin G was mainly due to reduced production of penicillinase in response to GE-AE. Given that the synchronous expression (Figure 9) of *agr* (*agrA* and *rnaIII*) and *blaZ* contradicts their negative transcriptional relationship (Figure 7), it is likely that *sarA* was primarily responsible for the down-regulation of penicillinase.

GE-AE targeting *sarA* and *sigB* downregulated multiple virulence factors involved in the initial and mature stages of biofilm formation. Among contributors observed in this study, *atlA*, *cidA*, *clfA*, *clfB*, *coa*, *crtN*, *ebpS*, *fnbA*, *hla*, *icaA*, *mecA*, and *vwb* can contribute *S. aureus* biofilm formation by directly or indirectly mediating surface adhesion, intercellular aggregation, or protecting bacterial cells from immune

clearance (Table S), (Table 4), which mechanistically involves various aspects of biofilm formation. The multi-targeted biofilm inhibitory property of GE-AE was also corroborated by significant reductions in cell adhesion (Figure 9 and 10), erythrocyte lysis (Figure 9, 11, and S4) and PIA production (Figure 9 and S5) in *S. aureus* ATCC43300 and clinical representatives. Certain flavonoids can effectively inhibit toxic expressions such as α -hemolysin (53). The down-regulation of *hld*, *psmA*, and *sfpA* suggests that inhibition of *S. aureus* ATCC43300 biofilm by GE-AE was independent of dissociation activity. Furthermore, *agr* (35, 41) was found to be associated with positive down-regulation of *crtN*, *hla*, and *mecA*, accounting for 25% of biofilm-positive contributors, while *sigB* and *sarA* were associated with 92% and 83% of biofilm-positive contributors, respectively (Table 4). Exceptionally, *mecA* (54) can in turn inhibit the activity of *agr* (Figure 7), thereby indirectly affecting the expression of related contributors. But judging from the synchronized expression of *mecA* and *agr* (Figure 9), the down-regulation of *mecA* was likely to be a concomitant phenomenon rather than a cause of the down-regulation of biofilm expression. It has been reported that extracellular DNA released from Atl-dependent autolysis is mainly responsible for the early stages of MRSA biofilm formation (55), whereas the FnBPs promote subsequent intercellular accumulation and biofilm maturation (56). This observation is consistent with the down-regulation of *atlA* and *fnbA* and accompanied biofilm attenuation in *S. aureus* ATCC43300 in response to GE-AE. Taking into account the regulatory relationship among *atlA*, *fnbA*, *sarA*, and *sigB*, the down-regulation of *atlA* is most likely due to reduced expression of *sigB*, while the down-regulation of *fnbA* is likely to be related to both *sigB* and *sarA*.

In conclusion, the synergistic antibacterial and inhibitory biofilm activity of *C. ambrosioides* L. alcohol extract combined with penicillin G against MRSA was

closely related to the interaction between main components of kaempferoside glycosides and quercetin. In mechanism, the increased sensitivity of MRSA to penicillin G was mainly related to the down-regulation of penicillinase expression with SarA as a potential drug target, while the biofilm inhibitory activity is mainly related to down regulation of various virulence factors involved in the initial and mature stages of biofilm development with SarA and/or σ B as potential drug target. This study provides a theoretical basis for further exploration of the medicinal activity of kaempferol rhamnosides and quercetin and its application in combination with the classic old drug penicillin G against MRSA biofilm infection.

AUTHOR CONTRIBUTIONS

X.H. analyzed the data and drafted the manuscript. W.Z. and Q.C. contributed to almost all data collection. Y.L. contributed to cell adhesion assay. G.B., T.L., J.B., and C.C. participated in the analysis of data. C.Y. contributed to the preparation of *C. ambrosioides* L. extracts. Y.Y., J.X., and Z.R. Y.J. and F.L. analyzed the data and revised the manuscript. All authors read and approved the final manuscript.

FUNDING

This study was supported by the National Nature Science Foundation of China (No. 82072297), the Open Project Program of Jiangsu Key Laboratory of Zoonosis (R1908 and R2109), the Six Talent Peaks Project in Jiangsu Province (2019-YY-065), and the High Level Talent Support Project of Yangzhou University.

TRANSPARENCY DECLARATIONS

None to declare.

SUPPLEMENTARY DATA

Supplementary data Table S1, Table S2, Table S3, Figure S1, Figure S2, Figure S3, Figure S4, and Figure S5 are available.

Reference

1. Tattavin P, Schwartz BS, Graber CJ, Volinski J, Bhukhen A, Bhukhen A, Mai TT, Vo NH, Dang DN, Phan TH, Basuino L, Perdreau-Remington F, Chambers HF, Diep BA. 2012. Concurrent epidemics of skin and soft tissue infection and bloodstream infection due to community-associated methicillin-resistant *Staphylococcus aureus*. Clin Infect Dis 55:781-8.
2. North EA, Christie R. 1946. Acquired resistance of staphylococci to the action of penicillin. Med J Aust 1:176-9.
3. Jevons MP. 1961. "Celbenin" - resistant Staphylococci. British Medical Journal 1:124-125.
4. Tong SY, Davis JS, Eichenberger E, Holland TL, Fowler VG, Jr. 2015. *Staphylococcus aureus* infections: epidemiology, pathophysiology, clinical manifestations, and management. Clin Microbiol Rev 28:603-61.
5. Chen F, Di H, Wang Y, Cao Q, Xu B, Zhang X, Yang N, Liu G, Yang CG, Xu Y, Jiang H, Lian F, Zhang N, Li J, Lan L. 2016. Small-molecule targeting of a diapophytoene desaturase inhibits *S. aureus* virulence. Nat Chem Biol 12:174-9.
6. Cavalla D, Singal C. 2012. Retrospective clinical analysis for drug rescue: for new indications or stratified patient groups. Drug Discov Today 17:104-9.
7. Costerton JW, Stewart PS, Greenberg EP. 1999. Bacterial biofilms: a common cause of persistent infections. Science 284:1318-1322.
8. Kumar A, Alam A, Rani M, Ehtesham NZ, Hasnain SE. 2017. Biofilms: Survival and defense strategy for pathogens. Int J Med Microbiol 307:481-489.
9. Moormeier DE, Bayles KW. 2017. *Staphylococcus aureus* biofilm: a complex

- developmental organism. *Mol Microbiol* 104:365-376.
10. Foster TJ. 2019. The MSCRAMM Family of Cell-Wall-Anchored Surface Proteins of Gram-Positive Cocci. *Trends Microbiol* 27:927-941.
 11. Cramton SE, Gerke C, Schnell NF, Nichols WW, Gotz F. 1999. The intercellular adhesion (*ica*) locus is present in *Staphylococcus aureus* and is required for biofilm formation. *Infect Immun* 67:5427-33.
 12. Qin Z, Ou Y, Yang L, Zhu Y, Tolker-Nielsen T, Molin S, Qu D. 2007. Role of autolysin-mediated DNA release in biofilm formation of *Staphylococcus epidermidis*. *Microbiology (Reading)* 153:2083-2092.
 13. Periasamy S, Joo HS, Duong AC, Bach TH, Tan VY, Chatterjee SS, Cheung GY, Otto M. 2012. How *Staphylococcus aureus* biofilms develop their characteristic structure. *Proc Natl Acad Sci U S A* 109:1281-6.
 14. Archer NK, Mazaitis MJ, Costerton JW, Leid JG, Powers ME, Shirtliff ME. 2011. *Staphylococcus aureus* biofilms: properties, regulation, and roles in human disease. *Virulence* 2:445-459.
 15. Gupta P, Kour J, Bakshi M, Kalsi R. 2022. Chapter 5 - Flavonoids, p 105-113. *In* Kour J, Nayik GA (ed), *Nutraceuticals and Health Care* doi:<https://doi.org/10.1016/B978-0-323-89779-2.00001-6>. Academic Press.
 16. Bouyahya A, El Omari N, El Menyiy N, Guaouguaou F-E, Balahbib A, Chamkhi I. 2022. Anti-Quorum Sensing Agents from Natural Sources, p 533-557. *In* Kumar V, Shriram V, Paul A, Thakur M (ed), *Antimicrobial Resistance: Underlying Mechanisms and Therapeutic Approaches* doi:10.1007/978-981-16-3120-7_17. Springer Nature Singapore, Singapore.
 17. Chu SS, Feng Hu J, Liu ZL. 2011. Composition of essential oil of Chinese *Chenopodium ambrosioides* and insecticidal activity against maize weevil,

Sitophilus zeamais. Pest Manag Sci 67:714-8.

18. Owolabi MS, Lajide L, Oladimeji MO, Setzer WN, Palazzo MC, Olowu RA, Ogundajo A. 2009. Volatile constituents and antibacterial screening of the essential oil of *Chenopodium ambrosioides* L. growing in Nigeria. Nat Prod Commun 4:989-92.
19. Kumar R, Mishra AK, Dubey NK, Tripathi YB. 2007. Evaluation of *Chenopodium ambrosioides* oil as a potential source of antifungal, antiaflatoxic and antioxidant activity. Int J Food Microbiol 115:159-64.
20. Monzote L, Garcia M, Pastor J, Gil L, Scull R, Maes L, Cos P, Gille L. 2014. Essential oil from *Chenopodium ambrosioides* and main components: activity against *Leishmania*, their mitochondria and other microorganisms. Exp Parasitol 136:20-6.
21. Harraz FM, Hammada HM, El Ghazouly MG, Farag MA, El-Aswad AF, Bassam SM. 2015. Chemical composition, antimicrobial and insecticidal activities of the essential oils of *Conyza linifolia* and *Chenopodium ambrosioides*. Nat Prod Res 29:879-82.
22. Genovese C, D'Angeli F, Bellia F, Distefano A, Spampinato M, Attanasio F, Nicolosi D, Di Salvatore V, Tempera G, Lo Furno D, Mannino G, Milardo F, Li Volti G. 2021. In Vitro Antibacterial, Anti-Adhesive and Anti-Biofilm Activities of *Krameria lappacea* (Dombey) Burdet & B.B. Simpson Root Extract against Methicillin-Resistant *Staphylococcus aureus* Strains. Antibiotics (Basel) 10.
23. He X, Yuan F, Lu F, Yin Y, Cao J. 2017. Vancomycin-induced biofilm formation by methicillin-resistant *Staphylococcus aureus* is associated with the secretion of membrane vesicles. Microb Pathog 110:225-231.

24. Li Y, Wang B, Lu F, Ahn J, Zhang W, Cai L, Xu J, Yin Y, Cao Q, Ren Z, He X. 2022. Synergistic Inhibitory Effect of Polymyxin B in Combination with Ceftazidime against Robust Biofilm Formed by *Acinetobacter baumannii* with Genetic Deficiency in AbaI/AbaR Quorum Sensing. *Microbiol Spectr* 10:e0176821.
25. Lewis RE, Diekema DJ, Messer SA, Pfaller MA, Klepser ME. 2002. Comparison of Etest, checkerboard dilution and time-kill studies for the detection of synergy or antagonism between antifungal agents tested against *Candida* species. *J Antimicrob Chemother* 49:345-51.
26. Tang J, Chen Y, Wang X, Ding Y, Sun X, Ni Z. 2020. Contribution of the AbaI/AbaR Quorum Sensing System to Resistance and Virulence of *Acinetobacter baumannii* Clinical Strains. *Infect Drug Resist* 13:4273-4281.
27. Gohar AA, Maatooq GT, Niwa M. 2000. Two flavonoid glycosides from *Chenopodium murale*. *Phytochemistry* 53:299-303.
28. He X, Li S, Yin Y, Xu J, Gong W, Li G, Qian L, Yin Y, He X, Guo T, Huang Y, Lu F, Cao J. 2019. Membrane Vesicles Are the Dominant Structural Components of Ceftazidime-Induced Biofilm Formation in an Oxacillin-Sensitive MRSA. *Frontiers in Microbiology* 10.
29. Ming D, Wang D, Cao F, Xiang H, Mu D, Cao J, Li B, Zhong L, Dong X, Zhong X, Wang L, Wang T. 2017. Kaempferol Inhibits the Primary Attachment Phase of Biofilm Formation in *Staphylococcus aureus*. *Front Microbiol* 8:2263.
30. Rachid S, Ohlsen K, Witte W, Hacker J, Ziebuhr W. 2000. Effect of subinhibitory antibiotic concentrations on polysaccharide intercellular adhesin expression in biofilm-forming *Staphylococcus epidermidis*. *Antimicrob Agents*

Chemother 44:3357-3363.

31. Mori A, Nishino C, Enoki N, Tawata S. 1987. Antibacterial activity and mode of action of plant flavonoids against *Proteus vulgaris* and *Staphylococcus aureus*. *Phytochemistry* 26:2231-2234.
32. Cho J, Kim H, Kim C, Cho S. 2017. Interaction with Polyphenols and Antibiotics. *Journal of Life Science* 27:476-481.
33. Batzilla CF, Rachid S, Engelmann S, Hecker M, Hacker J, Ziebuhr W. 2006. Impact of the accessory gene regulatory system (Agr) on extracellular proteins, *codY* expression and amino acid metabolism in *Staphylococcus epidermidis*. *Proteomics* 6:3602-13.
34. Arvidson S, Tegmark K. 2001. Regulation of virulence determinants in *Staphylococcus aureus*. *Int J Med Microbiol* 291:159-70.
35. Cheung GY, Wang R, Khan BA, Sturdevant DE, Otto M. 2011. Role of the accessory gene regulator agr in community-associated methicillin-resistant *Staphylococcus aureus* pathogenesis. *Infect Immun* 79:1927-35.
36. Jones RC, Deck J, Edmondson RD, Hart ME. 2008. Relative quantitative comparisons of the extracellular protein profiles of *Staphylococcus aureus* UAMS-1 and its *sarA*, *agr*, and *sarA agr* regulatory mutants using one-dimensional polyacrylamide gel electrophoresis and nanocapillary liquid chromatography coupled with tandem mass spectrometry. *J Bacteriol* 190:5265-78.
37. Xu L, Li H, Vuong C, Vadyvaloo V, Wang J, Yao Y, Otto M, Gao Q. 2006. Role of the *luxS* quorum-sensing system in biofilm formation and virulence of *Staphylococcus epidermidis*. *Infect Immun* 74:488-96.
38. Kong KF, Vuong C, Otto M. 2006. *Staphylococcus* quorum sensing in biofilm

- formation and infection. *Int J Med Microbiol* 296:133-9.
39. Xue T, Zhao L, Sun B. 2013. LuxS/AI-2 system is involved in antibiotic susceptibility and autolysis in *Staphylococcus aureus* NCTC 8325. *Int J Antimicrob Agents* 41:85-9.
 40. Cassat J, Dunman PM, Murphy E, Projan SJ, Beenken KE, Palm KJ, Yang SJ, Rice KC, Bayles KW, Smeltzer MS. 2006. Transcriptional profiling of a *Staphylococcus aureus* clinical isolate and its isogenic agr and sarA mutants reveals global differences in comparison to the laboratory strain RN6390. *Microbiology (Reading)* 152:3075-3090.
 41. Dunman PM, Murphy E, Haney S, Palacios D, Tucker-Kellogg G, Wu S, Brown EL, Zagursky RJ, Shlaes D, Projan SJ. 2001. Transcription profiling-based identification of *Staphylococcus aureus* genes regulated by the agr and/or sarA loci. *J Bacteriol* 183:7341-53.
 42. Schilcher K, Andreoni F, Dengler Haunreiter V, Seidl K, Hasse B, Zinkernagel AS. 2016. Modulation of *Staphylococcus aureus* Biofilm Matrix by Subinhibitory Concentrations of Clindamycin. *Antimicrob Agents Chemother* 60:5957-67.
 43. Bischoff M, Dunman P, Kormanec J, Macapagal D, Murphy E, Mounts W, Berger-Bachi B, Projan S. 2004. Microarray-based analysis of the *Staphylococcus aureus* σ B regulon. *J Bacteriol* 186:4085-99.
 44. Houston P, Rowe SE, Pozzi C, Waters EM, O'Gara JP. 2011. Essential role for the major autolysin in the fibronectin-binding protein-mediated *Staphylococcus aureus* biofilm phenotype. *Infect Immun* 79:1153-65.
 45. Rice KC, Patton T, Yang SJ, Dumoulin A, Bischoff M, Bayles KW. 2004. Transcription of the *Staphylococcus aureus* cid and lrg murein hydrolase

- regulators is affected by sigma factor B. *J Bacteriol* 186:3029-37.
46. Queck SY, Jameson-Lee M, Villaruz AE, Bach TH, Khan BA, Sturdevant DE, Ricklefs SM, Li M, Otto M. 2008. RNAIII-independent target gene control by the *agr* quorum-sensing system: insight into the evolution of virulence regulation in *Staphylococcus aureus*. *Mol Cell* 32:150-8.
 47. Bischoff M, Entenza JM, Giachino P. 2001. Influence of a functional *sigB* operon on the global regulators *sar* and *agr* in *Staphylococcus aureus*. *J Bacteriol* 183:5171-9.
 48. Lauderdale KJ, Boles BR, Cheung AL, Horswill AR. 2009. Interconnections between Sigma B, *agr*, and proteolytic activity in *Staphylococcus aureus* biofilm maturation. *Infect Immun* 77:1623-35.
 49. Pane-Farre J, Jonas B, Hardwick SW, Gronau K, Lewis RJ, Hecker M, Engelmann S. 2009. Role of RsbU in controlling SigB activity in *Staphylococcus aureus* following alkaline stress. *J Bacteriol* 191:2561-73.
 50. Rice KC, Firek BA, Nelson JB, Yang SJ, Patton TG, Bayles KW. 2003. The *Staphylococcus aureus cidAB* operon: evaluation of its role in regulation of murein hydrolase activity and penicillin tolerance. *J Bacteriol* 185:2635-43.
 51. Groicher KH, Firek BA, Fujimoto DF, Bayles KW. 2000. The *Staphylococcus aureus lrgAB* operon modulates murein hydrolase activity and penicillin tolerance. *J Bacteriol* 182:1794-801.
 52. Rice KC, Nelson JB, Patton TG, Yang SJ, Bayles KW. 2005. Acetic acid induces expression of the *Staphylococcus aureus cidABC* and *lrgAB* murein hydrolase regulator operons. *J Bacteriol* 187:813-21.
 53. Jiang L, Li H, Wang L, Song Z, Shi L, Li W, Deng X, Wang J. 2016. Isorhamnetin Attenuates *Staphylococcus aureus*-Induced Lung Cell Injury by

Inhibiting Alpha-Hemolysin Expression. *J Microbiol Biotechnol* 26:596-602.

54. Pozzi C, Waters EM, Rudkin JK, Schaeffer CR, Lohan AJ, Tong P, Loftus BJ, Pier GB, Fey PD, Massey RC, O'Gara JP. 2012. Methicillin resistance alters the biofilm phenotype and attenuates virulence in *Staphylococcus aureus* device-associated infections. *PLoS Pathog* 8:e1002626.
55. Zapotoczna M, McCarthy H, Rudkin JK, O'Gara JP, O'Neill E. 2015. An Essential Role for Coagulase in *Staphylococcus aureus* Biofilm Development Reveals New Therapeutic Possibilities for Device-Related Infections. *J Infect Dis* 212:1883-93.
56. O'Neill E, Pozzi C, Houston P, Humphreys H, Robinson DA, Loughman A, Foster TJ, O'Gara JP. 2008. A novel *Staphylococcus aureus* biofilm phenotype mediated by the fibronectin-binding proteins, FnBPA and FnBPB. *J Bacteriol* 190:3835-50.

Table 1 Compounds identified from alcohol extracts of *C. ambrosioides* L.

No.	RT (min)	MW (m/z)	Compounds	Relative proportion (%)	
				GE-AE	GE-AE-EA
1	19.013	582	Naringin dihydrochalcone ^a	0.00	3.72
2	20.018	302	Quercetin ^a	2.92	0.00
3	20.587	594	Kaempferol-3-rutinoside ^{a,b}	0.00	2.81
4	21.113	580	Naringin ^a	0.00	1.32
5	21.333	610	Rutin ^a	0.00	0.79
6	24.147	578	Kaempferol-3,7-dirhamnoside ^{a,b}	14.25	33.55
7	24.905	564	Kaempferol-3-O-apigenin-7-O-rhamnoside ^{a,b}	24.17	34.04
8	26.745	434	Kaempferol-7-O-rhamnoside ^{a,b}	2.69	0.00
9	31.861	418	Syringaresinol	29.59	1.13
10	33.578	606	Kaempferol-3-O-acetylapigenin-7-O-rhamnoside ^{a,b}	26.39	22.64

^a, flavonoid compounds; ^b, kaempferol glycosides; GE, the ear of *C. ambrosioides* L.

originated from Guangxi, China; AE, alcohol extraction; EA, fractional extraction

with ethyl acetate.

Table 2 Correlation matrix of Pearson correlation coefficients between compounds in biofilm inhibition activity against *S. aureus* BWSA11

Factor	PCC for ^b :			
	GE-AE	K-3,7-D	Quercetin	Rutin
GE-AE	1	.718*	0.461	0.474
K-3,7-D		1	-.954**	0.417
Quercetin			1	-0.414
Rutin				1

GE, the ear of *C. ambrosioides* L. originated from Guangxi, China; AE, alcohol extraction; EA, fractional extraction with ethyl acetate; K-3,7-D, kaempferol-3,7-dirhamnoside. PCCs were calculated for strain exposed to quercetin (0.49~15.63 µg/ml), GE-AE (0.98~500 µg/ml), KD (0.49~500 µg/ml), and rutin (1.95~500 µg/ml). **, significant at the level of 0.01 (2-tailed); *, significant at the level of 0.05 (2-tailed).

Table 3 Correlation matrix of Pearson correlation coefficients between compounds in antibacterial activity against *S. aureus* BWSA11

Factor	PCC for:			
	GE-AE	K-3,7-D	Quercetin	Rutin
GE-AE	1	.799**	.523	.836**
K-3,7-D		1	-.606	.495
Quercetin			1	.948**
Rutin				1

GE, the ear of *C. ambrosioides* L. originated from Guangxi, China; AE, alcohol extraction; EA, fractional extraction with ethyl acetate; K-3,7-D, kaempferol-3,7-dirhamnoside. PCCs were calculated for strain exposed to quercetin (0.49~125 µg/ml), GE-AE (0.98~500 µg/ml), KD (0.49~500 µg/ml), and rutin (1.95~500 µg/ml). **, significant at the level of 0.01 (2-tailed).

Table 4 Contribution of genes to penicillin resistance and biofilm formation and potential transcriptional regulation in *S. aureus* ATCC43300 in response to GE-AE

Genes	Penicillin resistance	Biofilm formation	Regulons			
			agr (25%)*	sarA (83%)	sigB (92%)	luxS
atIA	-	+		N	P	
blaZ	+		N	P		
cidA	-	+		N	P	
clfA		+	N	P	P	
clfB		+	N	P	P	
coa		+	N	P	P	
crtN		+	P	P	P	
ebpS		+	N	P	P	
fnbA		+	N	P	P	
icaA		+		P	P	N
hla		+	P	P	N	
hld		-	P	P	N	
lrgA	+	-	P	P	N	
mecA	+	+	P	P	P	
psma		-	P	P	N	
sspA		-	P	N	N	
vwb		+	N	P	P	

-, negative contributor; +, positive contributor; N, negative regulation; P, positive regulation; *, percentage of positively regulated contributors of biofilm formation.

Figure legends

Figure 1 Biofilm inhibitory activity of different extracts from *C. ambrosioides*

L. GE, the ear of *C. ambrosioides* L. originated from Guangxi, China; AE, alcohol extraction; EA, fractional extraction with ethyl acetate. All data are presented as means of three biological replicates.

Figure 2 Comparison of biofilm inhibitory activities between GE-AE and

representative components. GE, the ear of *C. ambrosioides* L. originated from Guangxi, China; AE, alcohol extraction; K-3,7-D, kaempferol-3,7-dirhamnoside. All data are presented as means of three biological replicates.

Figure 3 Heat map of anti-biofilm effects of quercetin in combination with

kaempferol glycoside. K-3,7-D, kaempferol-3,7-dirhamnoside. Not determined, the biofilm formation of the area that the bacterial inhibition rate is over 95% was not calculated. Black asterisks indicate significant decreases at a *P* value of 0.05 compared to control. Green asterisks indicate significant decreases at a *P* value of 0.05 compared to control and either of the two parallel single-agent groups. All data are presented as means of three biological replicates.

Figure 4 Comparison of bacterial inhibitory activities between GE-AE and

representative components. GE, the ear of *C. ambrosioides* L. originated from Guangxi, China; AE, alcohol extraction; K-3,7-D, kaempferol-3,7-dirhamnoside. All data are presented as means of three biological replicates.

Figure 5 Heat map of antibacterial effects of combinations between penicillin G

and representative components. GE, the ear of *C. ambrosioides* L. originated from Guangxi, China; AE, alcohol extraction; K-3,7-D, kaempferol-3,7-dirhamnoside. The area circled by the green line indicates that the bacterial inhibition rate is over 95%. All data are presented as means of three biological replicates.

Figure 6 Heat map of anti-biofilm effects of penicillin G in combination with GE-AE and representative components. GE, the ear of *C. ambrosioides* L. originated from Guangxi, China; AE, alcohol extraction; K-3,7-D, kaempferol-3,7-dirhamnoside. Not determined, the biofilm formation of the area that the bacterial inhibition rate is over 95% was not calculated. Black asterisks indicate significant decreases at a *P* value of 0.05 compared to control. Green asterisks indicate significant decreases at a *P* value of 0.05 compared to control and either of the two parallel single-agent groups. All data are presented as means of three biological replicates.

Figure 7 Network of transcriptional regulation of genes involved in *S. aureus* penicillin resistance and biofilm formation.

Figure 8 Heat map of gene expression of *S. aureus* ATCC43300 in response to GE-AE. GE, the ear of *C. ambrosioides* L. originated from Guangxi, China; AE, alcohol extraction. All data are presented as means of three biological replicates.

Figure 9 Matrix graph of correlation (Corr) among gene expression, biofilm formation, hemolysis, and cell adhesion in *S. aureus* ATCC43300 in response to GE-AE. GE, the ear of *C. ambrosioides* L. originated from Guangxi, China; AE,

alcohol extraction. Significant correlation was calculated at the level of $P < 0.05$.

Figure 10 Cell adhesion activity of *S. aureus* strains in the presence of GE-AE.

GE, the ear of *C. ambrosioides* L. originated from Guangxi, China; AE, alcohol extraction. All data are presented as means of three biological replicates.

Figure 11 Hemolytic activity of *S. aureus* strains in the presence of GE-AE.

GE, the ear of *C. ambrosioides* L. originated from Guangxi, China; AE, alcohol extraction. All data are presented as means of three biological replicates.

Responses to Reviewers

Re: Spectrum02782-22 (Global down-regulation of penicillin resistance and biofilm formation by MRSA is associated with the interaction between kaempferol rhamnosides and quercetin)

Authors: Xinlong He, et al.

We sincerely thank all Reviewers for your valuable comments and giving us the chance to revise our manuscript. We have finished the point-by-point responses for this manuscript, and you will find the changes in the Marked Up Manuscript up-loaded.

Responses to Reviewer #1 (Comments for the Author):

This study is well designed and written based on the results obtained. There are few minor comments.

1. In Table 1, add the concentration of each compound with relative proportions.

Response: Due to the inaccessibility of some standards, we did not calculate the actual concentrations of these compounds. Instead, we performed qualitative analysis on these compounds identified by MS and relative quantitative analysis based on their LC abundances. Since we do not know the actual concentrations, to make it clear that two extracts are comparable, we annotate the results of the difference analysis in Table 1 (Line 650-654) and describe in more detail where appropriate in the Results and Discussion sections (Line 278-283).

2. State the reason why the penicillin was used rather than other generations of beta-lactams or other classes of antibiotics.

Response: In fact, we tested five antibiotics, penicillin G, ceftazidime, erythromycin,

levofloxacin, and vancomycin, which belong to different generations of beta-lactams or other classes of antibiotics. It was found that only the combination of penicillin G and GE-EE produced a synergistic anti-biofilm effect against *S. aureus*, so the results showing were mainly related to penicillin G. A description of the relevant experimental design and experimental results has been made in the Abstract (Line 31-32), Introduction (Line 106-107), Materials and Methods (Line 174-177), and Results and Discussion (Line 337-339) sections.

3. Tables 2 and 3 should be combined.

Response: As suggested, Tables 2 and 3 have been combined (Line 656-662), and original Table 4 has been numbered as Table 3 (Line 383, 409, 419, and 663) subsequently.

4. Use "anti-biofilm activity" instead of biofilm inhibitory activity throughout the manuscript.

Response: As suggested, "biofilm inhibitory activity" has been changed to "anti-biofilm activity" throughout the manuscript including Supplemental Material.

Responses to Reviewer #3 (Comments for the Author):

1. More reliable result could be obtained if clinical samples increased and diverted.

Response: we agree that more reliable result could be obtained if clinical samples increased and diverted. In the next step, we will screen out a number of clinical *S. aureus* representative strains with different genetic backgrounds, test the universality

of the anti-biofilm activity of the combination of kaempferol and penicillin G, and further explore its molecular mechanism.

2. In methodology there is few grammar mistakes and better to avoid repetition.

Response: As suggested, all the grammar mistakes mentioned have been corrected as follow.

(1) "alcoholic extract" has been changed to "alcoholic extract" (Line 36);

(2) In Materials and Methods section, description of "CAEs were prepared by using immersion extraction (IE) and alcohol extraction (AE), respectively" has been changed to "CAEs were prepared by immersion extraction method using water and ethanol as the solvents" (Line 118-119). For ease of understanding, we have changed the abbreviation from the original "IE" (immersion extraction using water as solvent) to "WE", and the original "AE" (immersion extraction using ethanol as solvent) to "EE" throughout the manuscript including Supplemental Materials;

(3) "was" has been changed to "were" (Line 132).

3. Regulation of gene not discussed in details (the Method)

Response: As suggested, the molecular docking method has been described in more detail and reference has been cited (Line 245-261).

October 24, 2022

Dr. Feng Lu
Yangzhou University
Yangzhou
China

Re: Spectrum02782-22R1 (Global down-regulation of penicillin resistance and biofilm formation by MRSA is associated with the interaction between kaempferol rhamnosides and quercetin)

Dear Dr. Feng Lu:

Your manuscript has been accepted, and I am forwarding it to the ASM Journals Department for publication. You will be notified when your proofs are ready to be viewed.

Sincerely,

Cezar Khursigara
Editor, Microbiology Spectrum
